# Elevated Na is a dynamic and reversible modulator of mitochondrial metabolism in the heart

Yu Jin Chung[1], Zoe Hoare [1], Friedrich Baark[2], Chak Shun Yu [3], Jia Guo [4], William Fuller [5], Richard Southworth [2], Doerthe M. Katschinski [4], Michael P. Murphy [3], Thomas R. Eykyn [2] ✉ & Michael J. Shattock [1] ✉

Elevated intracellular sodium $Na_i$ adversely affects mitochondrial metabolism and is a common feature of heart failure. The reversibility of acute Na induced metabolic changes is evaluated in Langendorff perfused rat hearts using the Na/K ATPase inhibitor ouabain and the myosin-uncoupler para-amino-blebbistatin to maintain constant energetic demand. Elevated $Na_i$ decreases Gibb's free energy of ATP hydrolysis, increases the TCA cycle intermediates succinate and fumarate, decreases ETC activity at Complexes I, II and III, and causes a redox shift of CoQ to $CoQH_2$, which are all reversed on lowering $Na_i$ to baseline levels. Pseudo hypoxia and stabilization of HIF-1α is observed despite normal tissue oxygenation. Inhibition of mitochondrial Na/Ca-exchange with CGP-37517 or treatment with the mitochondrial ROS scavenger MitoQ prevents the metabolic alterations during $Na_i$ elevation. Elevated $Na_i$ plays a reversible role in the metabolic and functional changes and is a novel therapeutic target to correct metabolic dysfunction in heart failure.

Elevated myocardial Na (intracellular $[Na]_i$) is a hallmark of heart failure (HF), affecting numerous downstream pathways that govern cardiac function. In addition to its critical role in electrical excitability and contraction in cardiomyocytes, recent studies have demonstrated a link between Na and mitochondrial metabolism, via the mitochondrial Na/Ca exchanger (NCLX), whereby elevated cytosolic Na activates NCLX, decreasing mitochondrial Ca concentration[1–3]. A number of mitochondrial enzymes are Ca-sensitive including pyruvate dehydrogenase (PDH), isocitrate dehydrogenase (IDH) and α-ketoglutarate dehydrogenase (KDH). Recent work has also elucidated the Ca-sensitivities of several of the proteins in the electron transport chain (ETC) including all three proton pumps (Complex I, Complex III, and Complex IV)[4] and a potential role for Ca in the activation of the $FoF_1$-ATPase (Complex V, ATP synthase)[5]. Decreased mitochondrial Ca leads to reduced activity of these Ca-sensitive dehydrogenases as well as the ETC complexes, leading to impaired mitochondrial function and

metabolic inefficiency[6]. In turn, mitochondrial dysfunction can lead to the generation of reactive oxygen species (ROS) and subsequent oxidative stress which are widely reported to play a role in the aetiology and progression of HF[7].

A well-established metabolic phenotype in HF is a switch in substrate preference away from the preferred fatty acid (FA) oxidation towards a greater reliance on glucose oxidation, or glycolytic production of lactate. The phenomenon has been observed in humans[8,9] and various animal models of HF[10–12], including a model of HF induced by pressure overload in mice where intracellular Na is pathologically and chronically elevated[1]. This rerouting of substrate utilization is typically attributed to the energetic deficit that is thought to underlie contractile dysfunction, maladaptive cardiac remodelling, and progression of HF. The metabolic switch in substrate preference has also been observed in a transgenic mouse model of chronic Na elevation which lack an overt HF phenotype[1,13]. Similar alterations in substrate

[1]School of Cardiovascular and Metabolic Medicine and Sciences, King's College, London, UK. [2]School of Biomedical Engineering and Imaging Sciences, King's College London, London, UK. [3]MRC Mitochondrial Biology Unit and Department of Medicine, University of Cambridge, Cambridge, UK. [4]Institute of Cardiovascular Physiology, University Medical Centre, Göttingen, Germany. [5]School of Cardiovascular and Metabolic Health, College of Medical, Veterinary and Life Sciences, University of Glasgow, Glasgow, UK. ✉e-mail: thomas.eykyn@kcl.ac.uk; michael.shattock@kcl.ac.uk

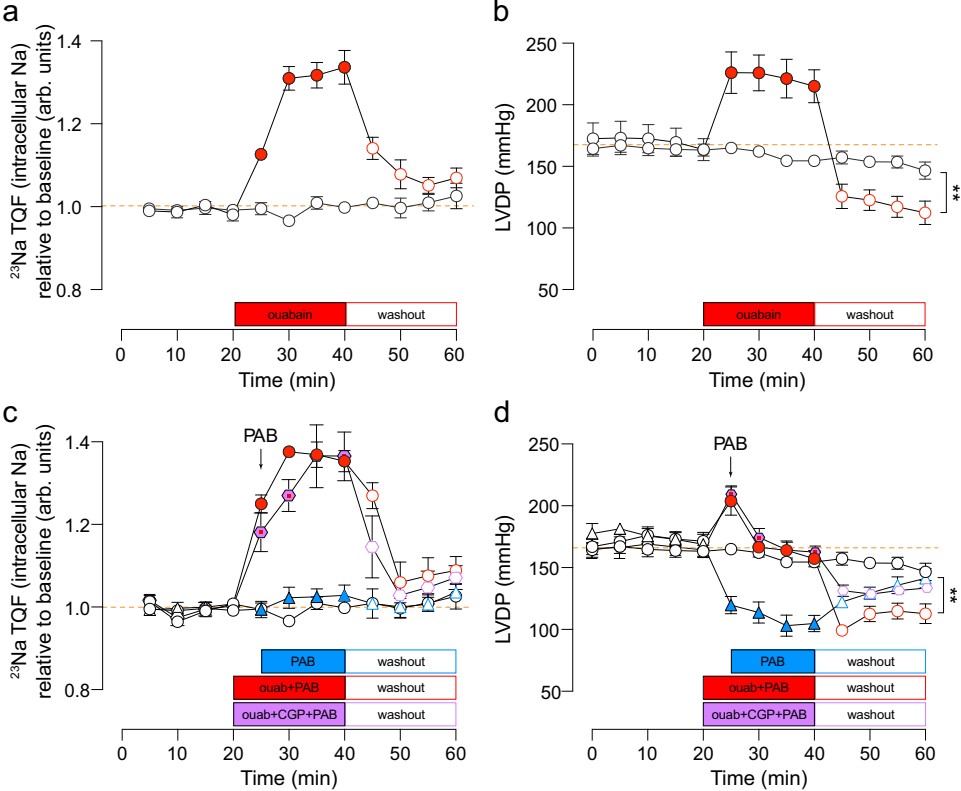

**Fig. 1 | Intracellular Na elevation and contractility in isolated Langendorff perfused rat hearts transiently exposed to 75 µM ouabain and 150 nM para-aminoblebbistatin (PAB). a** Time-courses of intracellular Na in the absence of PAB and **b** corresponding left ventricular developed pressure (LVDP) measured before (baseline), during ouabain treatment, and after washout. **c** Time-courses of intracellular Na and **d** corresponding left ventricular developed pressure (LVDP) measured before (baseline), during co-infusion of ouabain+PAB, ouabain+CGP + PAB, or PAB alone and after washout. Intracellular Na was measured using $^{23}$Na-TQF NMR spectroscopy. Para-aminoblebbistatin (PAB), used to remove the positive inotropy caused by elevation of Na, was titrated for 5 min following perfusion with ouabain and confirmation of positive inotropy; arrow denotes start of PAB titration in ouabain-treated hearts. White circle = time-matched control; red circle = ouabain treatment; purple diamond = ouabain + CGP blue triangle = PAB treatment only. $n = 6$ rats per condition; significance determined by two-tailed, unpaired student's t-test. **$P < 0.01$. Data plotted as mean ± SEM and source data are provided as a Source data file.

utilization can also be induced by acute elevation of [Na]$_i$ via pharmacological inhibition of the Na/K ATPase (NKA) with ouabain[1]. These observations suggest that the elevation of [Na]$_i$ alone is sufficient to cause these switches in substrate preference, independent of the presence of disease, thus implicating Na as a modulator of mitochondrial metabolism that underlies subsequent metabolic dysfunction that occur in HF. However, it is not known whether these metabolic alterations can be acutely reversed in HF by lowering Na back to healthy levels.

In the present work, we explore the mechanism by which acute [Na]$_i$ elevation affects mitochondrial metabolism and the reversibility of those alterations in an ex vivo model of acute Na elevation in Langendorff perfused rat hearts. We elucidate a mechanism by which elevated Na leads to decreased ATP supply, decreased ETC activity, a shift in redox balance leading to the generation of ROS and its pharmacological reversal with the NCLX inhibitor CGP-37157 and the mitochondria-targeted antioxidant MitoQ.

We conclude that altered cytosolic Na is a dynamic modulator of cardiac metabolism that is largely reversible and therefore could present a viable therapeutic target in HF.

## Results

### Intracellular Na concentration can be acutely and reversibly elevated in Langendorff perfused hearts

We first established a protocol whereby [Na]$_i$ can be acutely and reversibly elevated in Langendorff perfused hearts using the NKA inhibitor ouabain. Subsequent removal of ouabain from the perfusion buffer allowed re-activation of NKA which reduced Na back to baseline levels within 10 min (Fig. 1a). Na elevation causes positive inotropy through Ca loading (Fig. 1b), which results in an increase in metabolic demand due to increased contraction. Our objective was to interrogate the effect of [Na]$_i$ on metabolism, independent of contractility-related energetic demand. The increase in left ventricular pressure was therefore titrated back to baseline using the reversible contraction uncoupler para-aminoblebbistatin (PAB; a myosin II inhibitor derived from blebbistatin, Fig. 1d). This maintained hearts in a state of constant ATP demand during the period of Na elevation. We found that we could reliably elevate [Na]$_i$ and subsequently lower it to near-baseline levels by adding and removing ouabain and PAB from the perfusate. PAB did not attenuate the rise in [Na]$_i$ in response to ouabain (compare Figs. 1a and c). Importantly, both Na concentration and contractile function reached steady state within 20 min of drug application. Upon washout of ouabain, LVDP remained depressed compared to time-matched controls (Fig. 1d, open red circles). This reduction was not due to residual PAB, since the function of hearts treated with PAB alone recovered to time-matched controls within 20 min (Fig. 1d, compare open red circles vs open blue triangles).

### ATP reserve, ΔG$_{ATP}$ and creatine kinase activity are reduced by elevated intracellular Na

We next investigated whether acutely elevating [Na]$_i$ affects cardiac energetics using serially acquired $^{31}$P NMR spectroscopy. In ouabain-treated hearts, ATP levels were significantly reduced during the Na elevation period and further reduced at washout (Fig. 2a) compared to

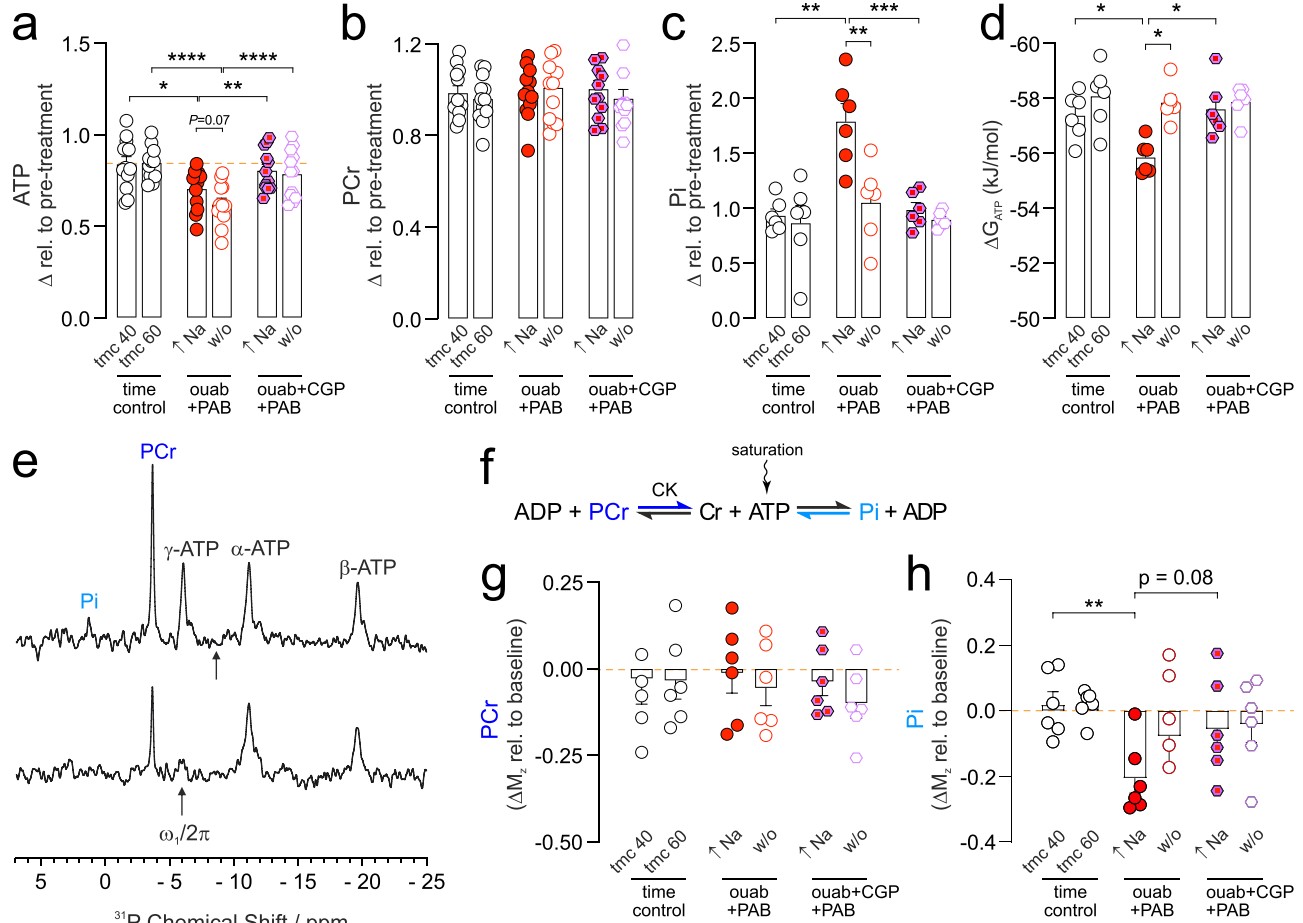

**Fig. 2 | $^{31}$P NMR cardiac energetics during intracellular Na elevation and washout. a** ATP, **b** phosphocreatine (PCr), **c** inorganic phosphate (Pi) and **d** free energy ($\Delta G_{ATP}$) of ATP hydrolysis at the end of the Na elevation (40 min) and at the end of the washout period (60 min) measured with $^{31}$P NMR spectroscopy. **e** Representative $^{31}$P saturation transfer NMR spectra where the arrows ($\omega_1/2\pi$) denote the frequency of the saturation pulse positioned midway between the α- and γ-ATP resonances (top spectrum, control) or positioned to saturate the γ-ATP peak (bottom spectrum), note the partial saturation of the PCr and Pi peaks due to

magnetization transfer. **f** Reaction scheme showing magnetization transfer in the direction of the coloured arrows. Quantification of the magnetization transfer from PCr to γ-ATP (**g**) and Pi to γ-ATP (**h**). Time control = time-matched control; ouab = ouabain; CK = creatine kinase. $n$ = 6 hearts or $n$ = 12 (**a**, **b**) in each group. Significance determined by two-tailed, unpaired student's t-test. *$P$ < 0.05, **$P$ < 0.01, ***$P$ < 0.001, ****$P$ < 0.0001. Data plotted as mean ± SEM and source data are provided as a Source data file.

time-matched controls. In contrast, PCr levels were unaffected by elevated Na and remained unchanged during washout (Fig. 2b). The decreased ATP level during high Na was accompanied by an increase in Pi (Fig. 2c) and a corresponding decrease in the free energy of ATP hydrolysis ($\Delta G_{ATP}$), which represents the driving force for all ATP-consuming processes (Fig. 2d). $\Delta G_{ATP}$ was restored to baseline upon washout despite a sustained reduction of ATP, which is a consequence of the conserved PCr levels. The decrease in ATP but not PCr under conditions of elevated Na is of interest since this is the opposite to what is widely observed in preclinical and clinical studies of HF and ischaemia-reperfusion, where PCr/ATP ratio is reduced in pathology, i.e. a decrease in the supply/demand balance due to increased demand would be accompanied by a decrease in PCr to maintain ATP levels constant[14–18]. Paradoxically, we observed the opposite, where ATP levels are decreased and PCr is maintained, despite a reduction in $\Delta G_{ATP}$ during elevated Na. The total adenine nucleotide pool was measured by $^1$H NMR and found not to be significantly altered during Na elevation (see Data Supplement Fig. S1).

Since $\Delta G_{ATP}$ and ATP levels are, in part, reflected by the thermodynamics of ATP hydrolysis and synthesis, and by the equilibrium position of the creatine kinase (CK) system, we investigated whether these reactions were affected during [Na]$_i$ elevation. The kinetics of the

CK reaction and ATP synthesis were measured using a $^{31}$P NMR saturation transfer protocol. This experiment measures the magnetisation transfer from PCr and Pi to γ-ATP when the γ-ATP peak is saturated (3 s duration in our experiments; Fig. 2e). The degree of saturation transfer is proportional to the CK flux for the conversion of PCr to ATP or to the rate of ATP synthesis from Pi, given by the product of the rate constants and their respective metabolite concentrations. We find that during Na elevation, the magnetisation transfer ($\Delta M_z$) from PCr to ATP was unchanged (Fig. 2f) while that from Pi to ATP was decreased (Fig. 2g), suggesting a decrease in the rate of ATP synthesis. We also assessed the net flux for ATP synthesis given by $\Delta M_z$ x [Pi] and found this to be similarly decreased by elevated Na (See Data Supplement Fig. S2). The decrease in the rate of ATP synthesis under conditions of Na elevation suggest alterations in the processes that generate ATP under conditions where ATP demand is constant and PCr is unchanged; these observations are consistent with a decrease in ATP supply.

## Elevation of intracellular Na reversibly increases glycolytic metabolism and reduces oxidative metabolism

Given that cardiac energetics were adversely affected by elevated [Na]$_i$ with a possible reduction in ATP supply measured by $^{31}$P NMR, we next

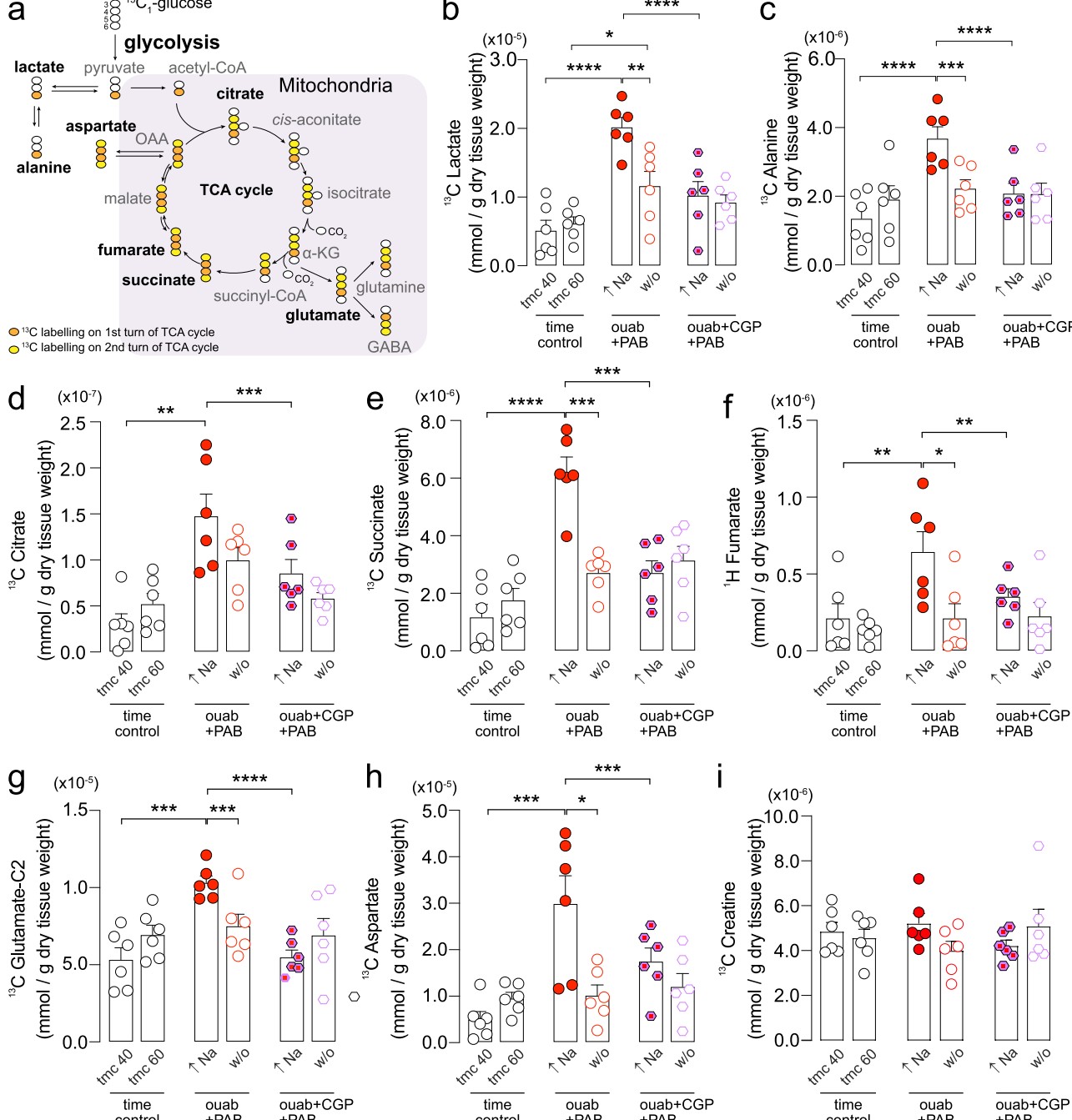

**Fig. 3 | Cardiac metabolomics during [Na]$_i$ elevation and washout. a** Schematic for $^{13}$C-labelling of cardiac metabolites arising from 1-$^{13}$C glucose. Orange circles correspond to the first round of labelling. Once $^{13}$C reaches succinate then it becomes scrambled in the 2 and 3 positions since the molecule is symmetric leading to a second round of labelling denoted by the yellow circles. OAA = oxaloacetic acid, α-KG = α-ketoglutarate. Quantification of **b** $^{13}$C lactate and **c** $^{13}$C alanine measured using $^{13}$C NMR spectroscopy. $n = 6$ hearts per groups. **d** $^{13}$C citrate, **e** $^{13}$C succinate, **f** $^{1}$H fumarate, **g** $^{13}$C glutamate C2, **h** $^{13}$C aspartate, and **i** $^{13}$C creatine measured using $^{13}$C or $^{1}$H NMR spectroscopy at the end of the Na elevation (40 min) and at the end of the washout period (60 min). Time control = time-matched control; ouab = ouabain-treated; CGP+ouab = CGP and ouabain co-treatment; w/o = washout. $n = 6$ hearts per group. Significance determined by two-tailed, unpaired student's t-test. *$P < 0.05$, **$P < 0.01$, ***$P < 0.001$, ****$P < 0.0001$. Data plotted as mean ± SEM and source data are provided as a Source data file.

sought to answer whether mitochondrial metabolism was also affected by changes in [Na]$_i$. Langendorff-perfused rat hearts were supplied with [1-$^{13}$C] glucose for 10 min during the steady-state period of Na elevation or after Na had returned to its baseline steady-state (Fig. 3a). The $^{13}$C labelling period corresponds to the times 30−40 min and 50−60 min in Fig. 1a. Hearts were subsequently snap-frozen, cardiac metabolites extracted and subjected to 1D $^{1}$H and 2D $^{1}$H/$^{13}$C HSQC NMR spectroscopy. Consistent with previous reports of increased glucose

utilization in chronic models of Na elevation in vivo[1], as well as in HF[19,20], acute elevation of [Na]$_i$ resulted in increased glycolytic metabolism, as evidenced by an increase in $^{13}$C glucose flux through lactate (Fig. 3b) and alanine (Fig. 3c), as well as increased incorporation of $^{13}$C into TCA cycle metabolites citrate, succinate, and fumarate (Fig. 3d−f), and increased incorporation into anaplerotic substrates glutamate and aspartate (Fig. 3g, h). In contrast, the concentration of creatine, a metabolite not derived from glucose metabolism, was not altered

(Fig. 3i). This suggests that the elevation of TCA intermediates likely resulted from their accumulation due to reduced activity of key enzymes in the TCA cycle and re-routing of metabolites into anaplerotic substrates such as glutamate and aspartate.

Upon washout of ouabain and restoration of $[Na]_i$, alanine concentration returned to baseline levels. Lactate concentration was significantly reduced during washout compared to Na elevation but remained elevated compared to time-matched controls. These results indicate that the increase in glycolytic metabolism during Na elevation is quickly restored when $[Na]_i$ is returned to baseline levels. Washout of ouabain and restoration of $[Na]_i$ also lowered the concentration of TCA intermediates back to levels comparable to time-matched controls showing that the effect on mitochondrial metabolism is also reversible (Fig. 3).

To further explore the effects of elevated $[Na]_i$ on mitochondrial metabolism, oxygen consumption was measured in isolated cardiac myocytes treated with ouabain. Elevated $[Na]_i$ resulted in a decrease in oxygen consumption in a dose-dependent manner (Fig. 4a). The effect of cytosolic $[Na]_i$ on mitochondrial respiration was also explored in permeabilised cardiomyocytes with increasing concentration of $[Na]$ (Fig. 4b–d). Saponin-treated cells were first treated with 0, 5, 10 or 20 mM NaCl. To stimulate mitochondrial respiration, the substrates K-malate and Na-pyruvate were added at 5 and 10 mM, respectively. As the Na-pyruvate contributed 10 mM Na to the reaction buffer, the final concentration of Na in each well was: 10 mM (control), 15, 20 or 30 mM (simulating the elevation of cytosolic $[Na]$). To initiate mitochondrial respiration, ADP was added to the cells in the presence of malate/pyruvate. Whereas variable cytosolic $[Na]_i$ did not affect oxygen consumption in quiescent mitochondria (Fig. 4b–d, data points between Na and mal/pyr), oxygen consumption was significantly reduced in a dose-dependent manner with rising $[Na]_i$ when respiration was initiated in the presence of ADP. These results are evidence of the effect of cytosolic $[Na_i]$ on modulating mitochondrial respiration and function.

The activities of ETC Complexes I, II and III were also measured from snap-frozen tissues under conditions of elevated Na and washout. The activities of all three ETC complexes were significantly decreased during Na elevation compared to time-matched controls. Upon washout of Na to physiological levels, the activity of Complex III was fully restored to levels comparable to time-matched controls while the activities of Complexes I and II were only partially restored (Fig. 4e–g).

## Inhibition of mitochondrial Na/Ca exchanger NCLX

It has previously been shown that elevated cytosolic $[Na]$ affects mitochondrial activity by decreasing mitochondrial $[Ca]_m$ via activation of the mitochondrial NCLX[1]. The above experiments are consistent with a decrease of $[Ca]_m$ that results in reduced activity of Ca-dependent mitochondrial enzymes including ATP synthase, Complex I, Complex III and the Ca-sensitive dehydrogenases pyruvate dehydrogenase, isocitrate dehydrogenase and α-ketoglutarate dehydrogenase[21,22]. Thus, preventing the decrease in $[Ca]_m$ by inhibiting Ca efflux from the mitochondria should prevent the metabolic alterations even when $[Na]_i$ is elevated. When hearts were co-treated with the NCLX inhibitor CGP-37157 and ouabain (CGP+ouab), LV dysfunction was largely prevented during washout compared to hearts treated only with ouabain (Fig. 1d). Decreased ATP and Pi levels, reduction in $\Delta G_{ATP}$, and reduced CK activity during Na elevation were also prevented by CGP (Fig. 2). Similarly, the elevation of lactate, alanine (Fig. 3b, c) and TCA intermediates (Fig. 3d–h) observed during Na elevation were also prevented by CGP co-treatment. Furthermore, the reduced oxygen consumption observed during Na elevation in isolated cardiomyocytes, either by NKA inhibition with ouabain or directly by increasing cytosolic Na, was also prevented by CGP (Fig. 4a, d). Taken together, these results confirm the link between

$[Na]_i$ and $Ca_m$ via NCLX and underscore the critical role that cytosolic Na plays in modulating mitochondrial activity.

## Elevated Ca alone without an increased Na does not cause the metabolic changes

Increased cytosolic $[Na]$ leads to a concomitant increase in cytosolic Ca via sarcolemmal NCX[23]. To exclude the possibility that the metabolic changes observed under conditions of elevated Na were driven by elevated Ca, cytosolic $[Ca]$ was raised in the myocardium by perfusing hearts in high Ca Krebs buffer (3.5 mM). This concentration of Ca in the perfusate resulted in a similar inotropy as that observed with Na elevation, but without altering intracellular $[Na]_i$ (Fig. 5a, b). Hearts were similarly treated with PAB to remove the effect of Ca-driven inotropy and maintain a constant metabolic demand throughout the protocol (Fig. 5a). Under these conditions, forward mode NCLX would not be activated and, in contrast to the combined Na and Ca elevation (as in ouabain), $[Ca]_m$ would be maintained or even raised[24].

Elevated cytosolic Ca alone, in the absence of altered contractility, did not affect cardiac energetics in terms of ATP and PCr levels (Fig. 5c, d). This contrasts with the decreased ATP levels observed during Na elevation (Fig. 2A-B). A modest decrease in Pi was noted in Ca elevation, but this was not statistically significant (Fig. 5e). The $\Delta G_{ATP}$ of ATP hydrolysis also remained unchanged by Ca elevation when demand was maintained constant (Fig. 5f), whereas during Na elevation, the magnitude of $\Delta G_{ATP}$ was significantly lower compared to controls (Fig. 2D).

To further understand the metabolic effects of Ca elevation alone on myocardial metabolism, cardiac metabolites were analysed using $^1$H NMR spectroscopy. Lactate, succinate, fumarate, and glutamate concentrations remained unchanged by elevated cytosolic Ca (Fig. 5g, i–k); interestingly lactate concentration was significantly reduced in Ca elevation compared to controls (Fig. 5g). This metabolic profile is in stark contrast to Na elevation, where these metabolites were significantly elevated in response to increased $[Na]_i$ (Fig. 3). Thus, the metabolic changes observed during Na elevation cannot be attributed to elevated cytosolic Ca secondary to cytosolic Na elevation, but rather are a Na effect.

## Na elevation enhances the reduction of the CoQ pool and myocardial ROS production

The decreased ETC activity observed in Fig. 4e–g, particularly the reported Ca-sensitivity of Complex III, could result in reduction of the Coenzyme Q (CoQ) pool, leading to decreased activity of Complex II which is not itself reported to be Ca-sensitive. We measured the CoQ redox state in hearts at the end of the Na elevation and after washout and found a significant shift towards the reduced $CoQH_2$ state over the oxidised CoQ state during Na elevation, and this was reversed with ouabain washout (Fig. 6).

A decrease in $CoQH_2$ oxidation by Complex III would cause a shift in the redox balance towards $CoQH_2$ (the reduced form) which provides possible a mechanism for the enhanced generation of mitochondrial reactive oxygen species (ROS) either through reverse electron transport (RET) at Complexes I, or by stabilisation of a CoQ semiquinone at Complex III during Na elevation. Kohlhaas and colleagues recently showed that elevated cytosolic $[Na]$ increases mitochondrial formation of ROS in failing cardiomyocytes[7]. We determined whether Na elevation resulted in elevated ROS production using isolated primary cardiomyocytes loaded with the mitochondrial ROS indicator MitoSOX. Upon elevation of $[Na]_i$ with ouabain, the rate of MitoSOX oxidation increased significantly compared to time-matched controls in a dose-dependent manner (Fig. 6c, d). Importantly this occurred immediately on ouabain treatment and not subsequent to the metabolic changes that we observed. The rate of MitoSOX oxidation was not significantly further increased upon ouabain washout

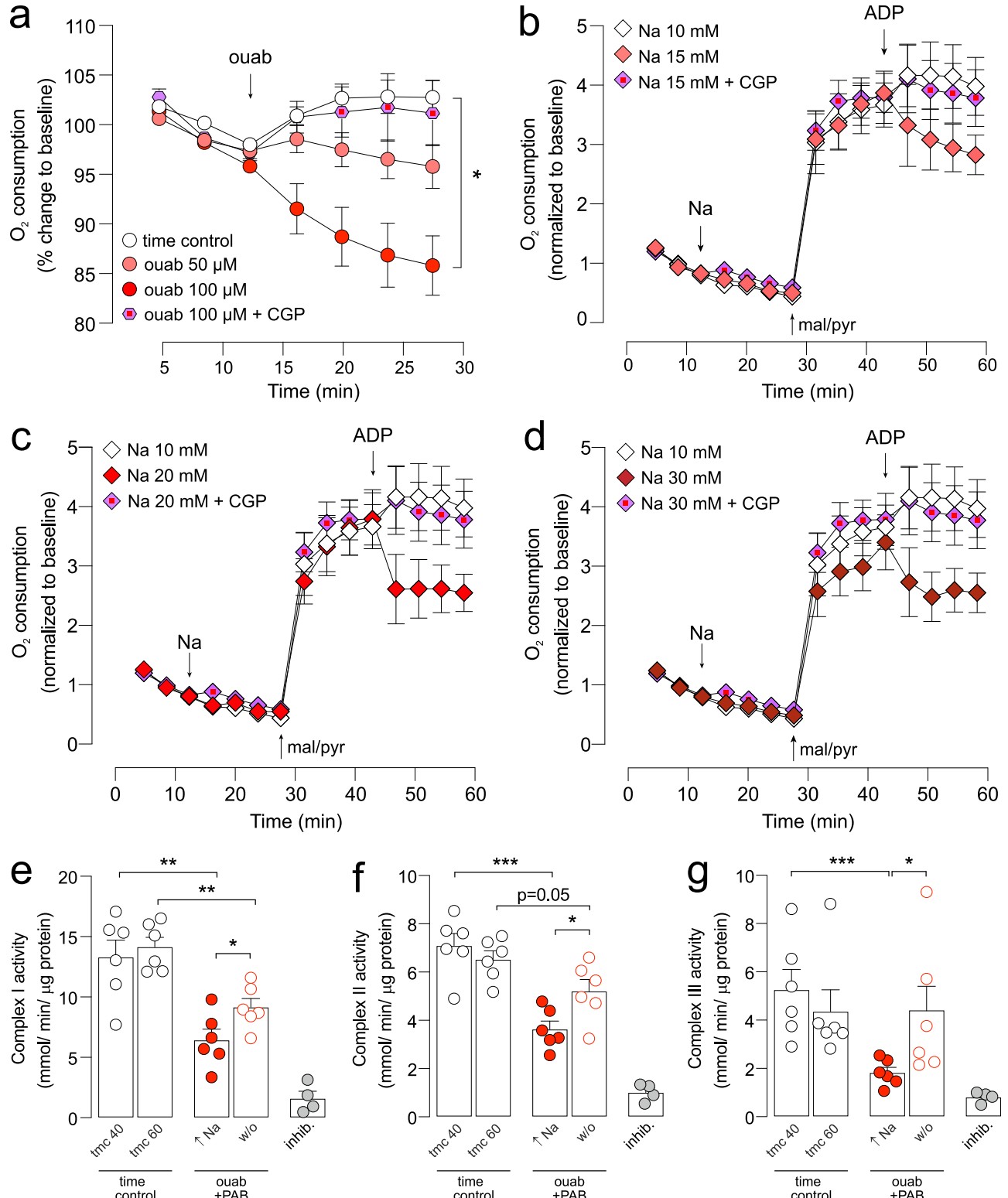

**Fig. 4 | Effect of [Na]$_i$ elevation on mitochondrial oxidative phosphorylation.**
Oxygen consumption was measured in isolated rat cardiomyocytes using the
Seahorse XFe24 platform. **a** Oxygen consumption rate (OCR) measured in
intact cardiomyocytes treated with 50 mM ouabain, 100 mM ouabain, or 100 mM
ouabain + CGP plotted as % change relative baseline. **b**–**d** Oxygen consumption rate
(OCR) measured in permeabilised cardiomyocytes supplemented with different
buffer Na concentrations, **b** 10 vs 15 mM, **c** 10 vs 20 mM or **d** 10 vs 30 mM. Also
shown is 30 mM + CGP treatment. Control Na concentration was 10 mM. Data
normalized to baseline. For (**a**–**d**), $n = 4$ rats, $N = 4$ technical replicates per rat per
condition. Significance determined by nested (hierarchical) one-way ANOVA.

*$P < 0.05$. **e**–**g** Electron transport chain activities of **e** Complex I, **f** Complex II and
**g** Complex III, measured at the end of the Na elevation (40 min) and at the end of
the washout period (60 min) for time-matched control (white data points) vs
ouabain treated (high Na, filled red data points) vs washout (open red data points);
inhib = Complex inhibitor rotenone for **e**, malonate for **f** and antimycin A for **g**. $n = 6$
hearts per group; $n = 4$ hearts for inhibitor. Significance determined by two-tailed,
unpaired student's t-test. Significance determined by nested (hierarchical) one-way
ANOVA. *$P < 0.05$, **$P < 0.01$, ***$P < 0.001$. Data plotted as mean ± SEM and source
data are provided as a Source data file.

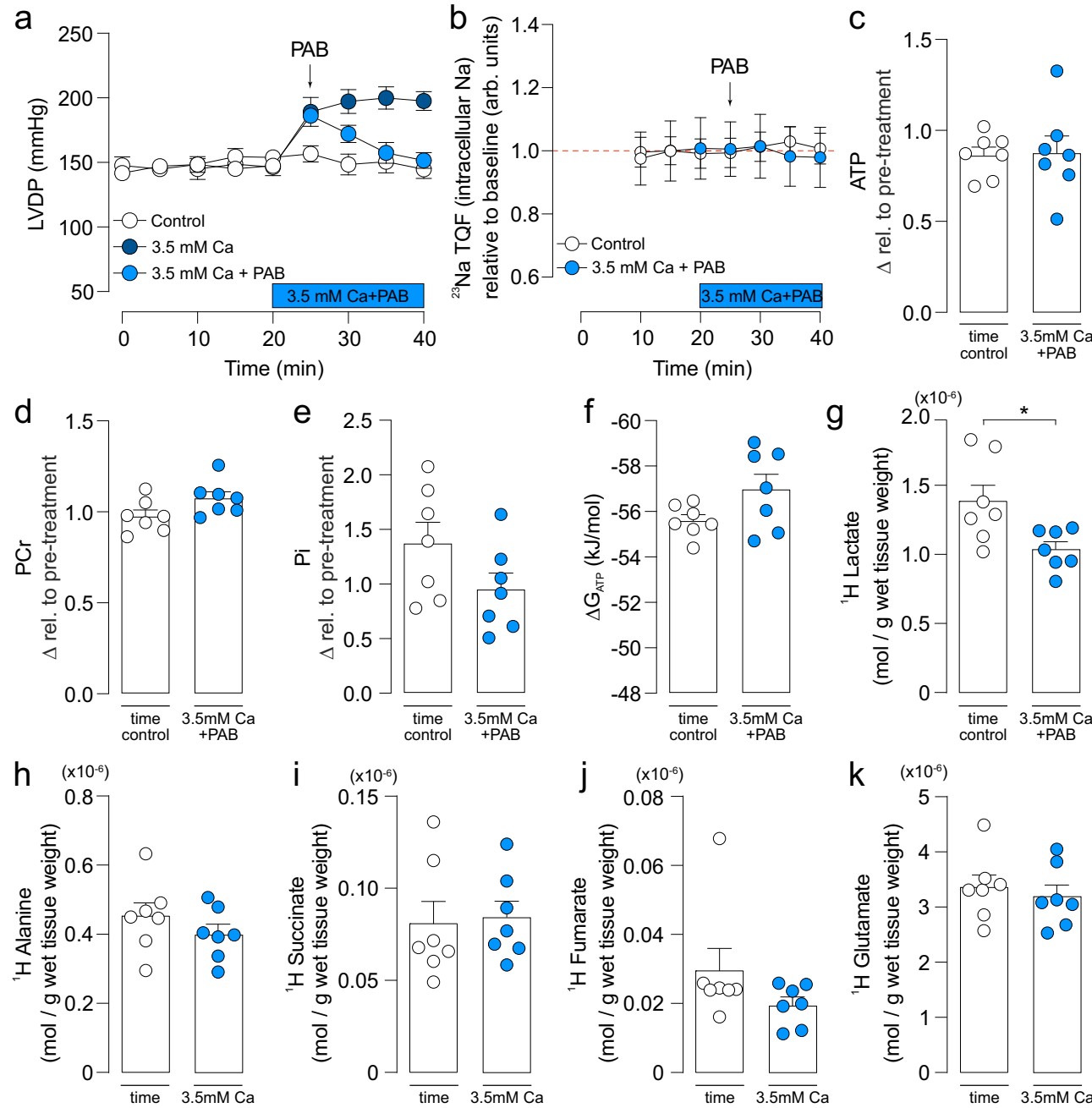

**Fig. 5 | Effect of [Ca]i elevation cardiac function and metabolomics. a** Time-course of left ventricular developed pressure (LVDP) in high Ca buffer (3.5 mM CaCl₂, dark blue filled symbols) and high Ca buffer titrated with para-aminoblebbistatin (3.5 mM CaCl₂ + PAB, light blue filled symbols). **b** Intracellular Na measured by $^{23}$Na-TQF NMR is unchanged during perfusion with 3.5 mM CaCl₂ + PAB. **c** ATP, **d** phosphocreatine (PCr), **e** inorganic phosphate (Pi) and **f** free energy ($\Delta G_{ATP}$) of ATP hydrolysis measured after perfusion with 3.5 mM CaCl₂ + PAB showing unaltered cardiac energetics. Metabolite concentration measured for control and high Ca buffer + PAB: **g** lactate, **h** alanine, **i** succinate, **j** fumarate and **k** glutamate using $^{1}$H NMR spectroscopy. $n = 7$ hearts per groups. Significance determined by two-tailed, unpaired student's t-test. ****$P < 0.0001$. Data plotted as mean ± SEM and source data are provided as a Source data file.

suggesting that the generation of ROS during elevated Na is mechanistically different to that observed during reperfusion injury[25].

Finally, we hypothesised that a decrease in the CoQ pool due to reduced Complex III activity could lead to elevated ROS that might be mitigated by supplementing the hearts with the mitochondria-targeted antioxidant Mitoquinone (MitoQ). Rats were fed MitoQ for 2 weeks and their hearts were then subject to the same Langendorff perfusion protocol with Na elevation and washout. Functionally, MitoQ-treated hearts recovered to near-baseline levels after 20 min

of washout following Na elevation and significantly better than sham hearts subject to the same protocol (Fig. 6e). During the Na elevation period, MitoQ-treated hearts had significantly higher ATP levels during ouabain treatment compared to sham (Fig. 6f) and their $\Delta G_{ATP}$ was also significantly higher than sham hearts (Fig. 6g). Furthermore, MitoQ-treated hearts had significantly lower concentrations of the glycolytic products lactate and alanine (Fig. 6h, i), as well as lower succinate and fumarate concentrations during ouabain treatment (Fig. 6j, k) compared to sham. MitoQ pre-treatment also

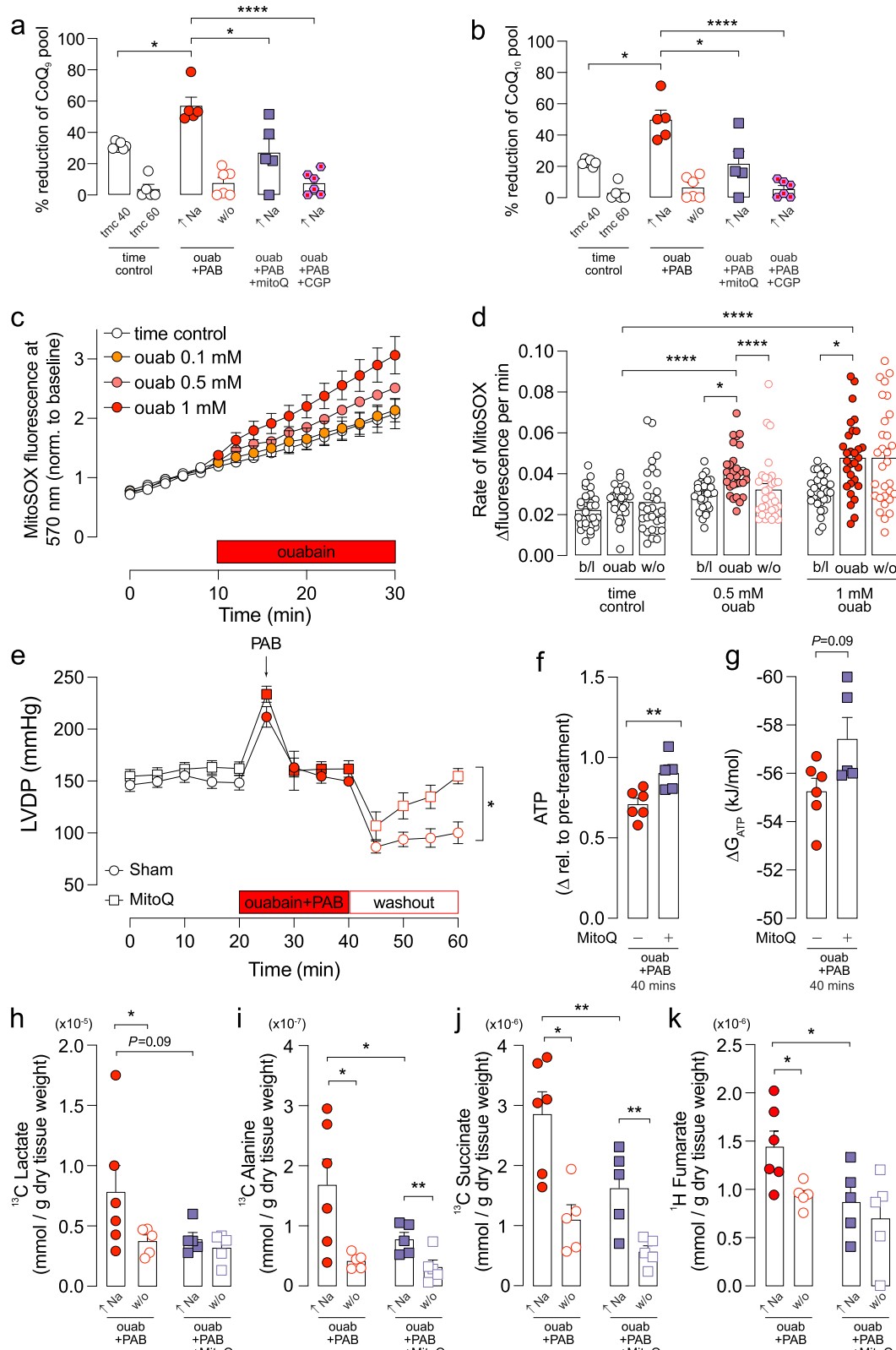

reversed the shift in redox balance back towards CoQ, the oxidised form (Fig. 6a, b).

## Na elevation elicits pseudohypoxia

The elevation of succinate and fumarate during Na elevation (Fig. 3e, f) are of particular interest since these are known to act as oncometabolites which inhibit the activities of 2-oxoglutarate-dependent dioxygenases such as the prolyl hydroxylase domain (PHD) enzymes[26,27]. PHDs are regulators of the hypoxia inducible factor-1α (HIF-1α), which in turn is a key mediator of glycolytic activity in cells and a transcription factor controlling the expression of a number of target enzymes. We therefore sought to answer whether the accumulation of succinate and fumarate following acute Na elevation was sufficient to induce the stabilization of HIF-1α, independent of

**Fig. 6 | [Na]ᵢ elevation reduces CoQ pool and causes ROS production.** Percentage of reduced $CoQH_2$ over total CoQ ($CoQ+CoQH_2$) pool for **a** $CoQ_9$ and **b** $CoQ_{10}$ in myocardium measured at the end of the Na elevation (40 min) and at the end of the washout period (60 min). $n = 5$ rats per group ($n = 6$ for ouabain+PAB + CGP). **c** Time course of ROS production in isolated mouse ventricular cardiomyocytes measured as an increase in the fluorescence of the ROS reporter MitoSOX at 570 nm in response to increasing concentration of ouabain. **d** Rate of change of superoxide accumulation measured in MitoSOX-loaded mouse cardiomyocytes subject to Na elevation via ouabain and ouabain washout; values normalized to baseline. $N = 30$ cells from $n = 3$ mice per group; $N = 5$ cells from $n = 1$ mouse for 0.1 mM ouabain group. **e–k** MitoQ rats were fed 500 µM of MitoQ in drinking water

for 2 weeks and their cardiac metabolism compared against sham rats on normal drinking water. **e** Time-course of left ventricular developed pressure (LVDP) in the hearts of sham vs MitoQ hearts with ouabain-induced Na elevation and washout period. **f** ATP and **g** $\Delta G_{ATP}$ during Na elevation in sham and MitoQ-treated hearts. Concentration of **h** $^{13}C$ lactate, **i** $^{13}C$ alanine, **j** $^{13}C$ succinate, and **k** $^{1}H$ fumarate in sham vs MitoQ hearts during Na elevation and washout (w/o). $n = 5$ (for sham) and $n = 6$ hearts per group. Significance determined by nested (hierarchical) one-way ANOVA for **d** two-tailed, unpaired student's t-test for (**a**, **b**) and (**e–g**) two-way ANOVA with post-hoc Sidak's multiple comparisons test (**h–k**); *$P < 0.05$, **$P < 0.01$, ****$P < 0.0001$. Source data are provided as a Source data file.

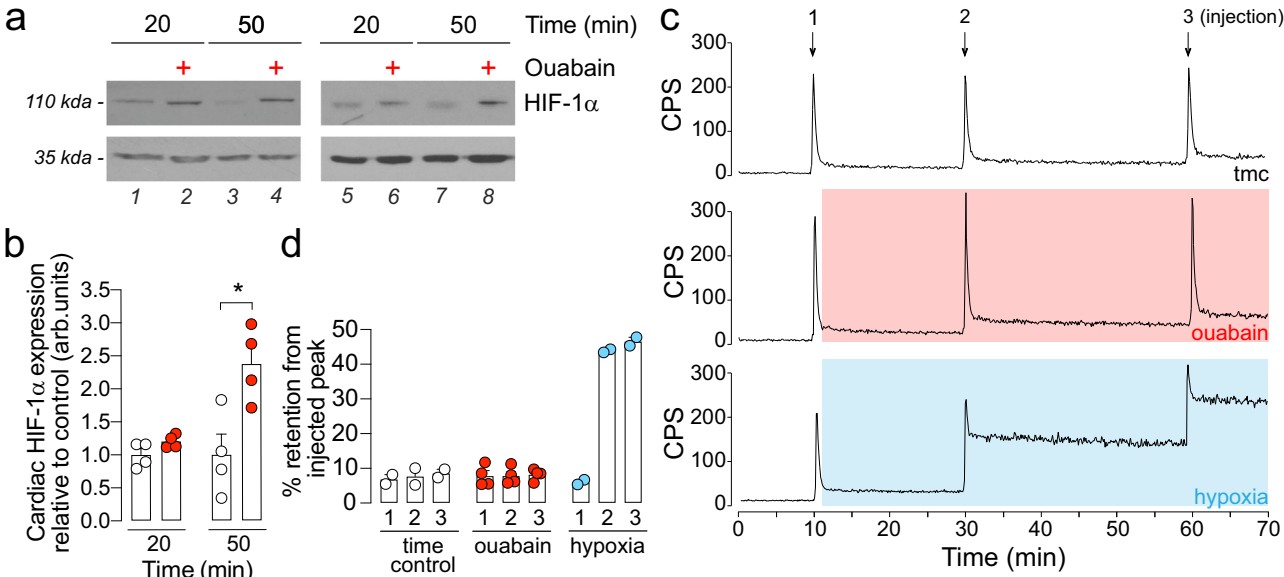

**Fig. 7 | Acute [Na]ᵢ elevation causes pseudohypoxia in hearts. a** HIF-1α protein expression measured by Western blotting and normalised to GAPDH loading control for control hearts and hearts treated with ouabain (high Na) for 20 min and 50 min. **b** Quantification of band density using ImageJ for control and ouabain treated hearts. $n = 4$ hearts per group. Data plotted as mean ± SEM. Significance determined by two-tailed, unpaired student's t-test; *$P > 0.05$. **c** Characterisation of tissue hypoxia by monitoring the cardiac accumulation of the hypoxia selective

radiotracer $^{64}Cu$-CTS by real-time γ-detection. Radioactivity measured in counts per second (CPS). Numbered arrows denote time-point and number of injections of radiotracer. 0% $O_2$ used as positive control for hypoxia. $n = 4$ hearts for ouabain-treated group, $n = 2$ for time-matched control (time control) and positive control (hypoxia). **d** Baseline corrected $^{64}Cu$ retention in time-matched control, ouabain treated and hypoxic hearts calculated as percentage of injected peak. Data plotted as mean ± SEM and source data are provided as a Source data file.

decreased tissue oxygenation (hypoxia). HIF-1α protein level was increased by 20% ($p = 0.15$) and by 230% ($p = 0.02$) after 20 min and 50 min duration of [Na]ᵢ elevation, respectively (Fig. 7a, b). To determine whether the observed HIF-1α stabilisation was due to Na per se or to possible tissue hypoxia, hearts were perfused with the PET tracer $^{64}Cu$-2,3-pentanedione bis(thiosemi-carbazone ($^{64}Cu$-CTS) during the period of [Na]ᵢ elevation with ouabain. These PET imaging agents are highly selective for tissue hypoxia, with $^{64}Cu$-CTS specifically accumulating in cardiac tissue at the critical hypoxic threshold where the heart becomes energetically compromised and HIF-1α becomes stabilised due to low $O_2$[28–30]. In this study however, [Na]ᵢ elevation caused no such cardiac accumulation of $^{64}Cu$-CTS (Fig. 7c, d), suggesting that elevated [Na]ᵢ causes HIF-1α stabilization by invoking a pseudohypoxic state independent of tissue oxygenation.

## Discussion

The present study shows, for the first time, a reversible role of intracellular Na in modulating cardiac energetics, mitochondrial redox state, and mitochondrial function in the intact, otherwise healthy myocardium. Acute elevation of [Na]ᵢ alone is sufficient to cause changes in cardiac energetics and metabolism which are largely reversible in the acute setting. These findings describe a novel mechanism through which [Na]ᵢ dynamically modulates mitochondrial

activity in the myocardium, causing changes which precede the energetic failure that characterises HF pathology, thus implicating Na elevation as an early event in the pathophysiology of HF and a potential driver of disease progression.

The proposed mechanisms supported by the data in this study are depicted in Fig. 8. The sequence of events in our proposed mechanism are as follows. The ex vivo model of acute Na elevation used in this study involved elevating [Na]ᵢ to pathological levels in Langendorff-perfused rat hearts by inhibiting NKA with ouabain, followed by its removal from the perfusate to allow reactivation of NKA and extrusion of excess Na to return to baseline levels. Contractility was kept constant by co-perfusing with the reversible myosin II inhibitor para-aminoblebbistatin (compare Fig. 1b and d) to partially uncouple contraction and remove the inotropic effect caused by Na elevation and subsequent Ca loading. This ensured that any observed energetic and metabolic changes could be attributed to alterations in [Na]ᵢ, independent of changes in contractility-linked energetic demand. We therefore confirmed that ouabain acts a reversible inhibitor of NKA and that [Na]ᵢ can be elevated and subsequently brought back down to near-baseline levels on washout (Fig. 1a) while PAB did not affect Na loading (Fig. 1c).

A direct link between cytoplasmic Na and mitochondrial function exists via the communication through mitochondrial NCLX, which

## a  Metabolic effects of cytoplasmic Na elevation

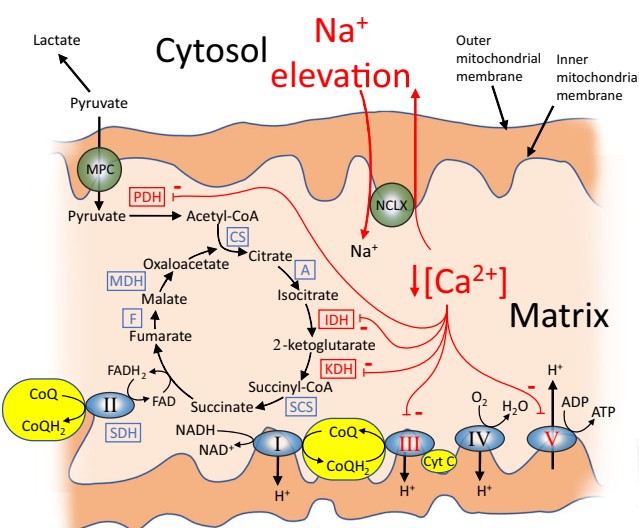

## b  ROS generation and pseudohypoxia

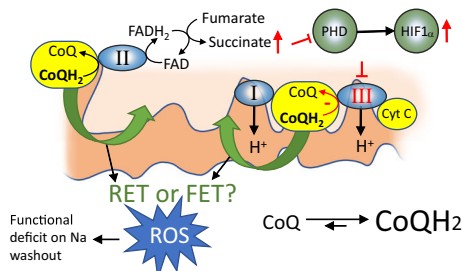

## c  Mechanism of action of MitoQ

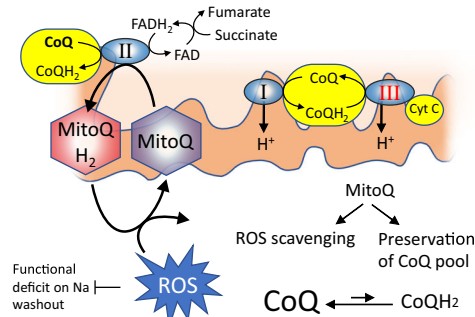

## d  Summary of the experimental observations

| Observation | Explanation |
| --- | --- |
| a - Lactate and alanine increased | Pyruvate entry into TCA cycle inhibited at PDH. Elevated cytosolic Ca increases ATP demand by Ca ATPases such as SERCA. |
| a - Citrate elevated | IDH inhibited. |
| a - Glutamate elevated | KDH inhibited. |
| a - Succinate elevated | Complex II and CoQ recycling inhibited or reverse electron transport at Complex II reduces fumarate to succinate. |
| a - ATP reduced | Complex V ($F_1F_0$ ATPase) activity decreased. |
| b - HIF1– stabilised under normoxia | Elevated succinate inhibits PHDs which reduces degradation of HIF1Xo |
| b – Increased ROS production | ROS produced either by FET or RET at Complexes I and II. ROS contributes to functional deficit on Na washout. |
| c - Effect of mitoQ | MitoQ can access the active site of Complex II (but not Complexes I and III). MitoQ is reduced to $MitoQH_2$ by SDH and is then is re-oxidised back to MitoQ by ROS. MitoQ therefore preserves the CoQ pool, prevents succinate accumulation and scavenges ROS. MitoQ prevents the functional deficit on Na washout. |

**Fig. 8 | Proposed sequence of events leading to altered mitochondrial metabolism by elevated intracellular Na. a** Elevated cytoplasmic Na leads to activation of mitochondrial NCLX leading to decreased Ca in the mitochondrial matrix and reduced activity of Ca-sensitive mitochondrial enzymes including pyruvate dehydrogenase (PDH), isocitrate dehydrogenase (IDH), α-ketoglutarate dehydrogenase (KDH) and Complex III and V (ATP synthase). Ca-sensitive enzymes are indicated in red while Ca-insensitive enzymes are blue. The normal flow of electrons through the ETC complexes are accepted by CoQ to produce $CoQH_2$ at Complexes I and II. $CoQH_2$ is reoxidised back to CoQ at Complex III allowing electron transport to continue at Complexes I and II. **b** Under conditions of high cytoplasmic Na, matrix Ca will be decreased leading to reduced activity of Complex III, decreased recycling of CoQ and a build-up of $CoQH_2$. Excess electrons in the $CoQH_2$ pool can lead to ROS generation via forward electron transport (FET) or reverse electron transport (RET) at Complexes I and II. There are two possible mechanisms for the build-up of succinate, (i) reduced CoQ availability will decrease activity of Complex II resulting

in decreased oxidation of succinate to fumarate, or (ii) RET at Complex II could reduce fumarate back to succinate in the reverse direction. The build-up of succinate will inhibit proly hydroxylases (PHDs) causing decreased degradation of HIF1α. **c** Pre-treatment in vivo with MitoQ loads the mitochondria with an exogenous antioxidant that is an analogue of CoQ. This could buffer the CoQ pool at Complex II by reducing MitoQ to $MitoQH_2$, limiting the build-up of succinate. $MitoQH_2$ is recycled to MitoQ by scavenging ROS, preventing the ROS-induced functional deficit on ouabain washout. *Note:* MitoQ can access the active site of Complex II but not Complexes I and III. Additional abbreviations: aconitase (A); citrate synthase (CS); cytochrome C (Cyt C); ubiquinone (CoQ); ubiquinol ($CoQH_2$); fumarase (F); malate dehydrogenase (MDH); mitochondrial pyruvate carrier (MPC); mitochondrial Na/Ca exchanger (NCLX); succinyl CoA synthetase (SCS); succinate dehydrogenase (SDH). Basic layout redrawn from Williams et al. (2015). Panel **d** links the observations made in this study to this model[22].

becomes activated under conditions of elevated cytoplasmic [Na], resulting in decreased mitochondrial calcium $[Ca]_m$ (Fig. 8a). This leads to reduced activities of the various Ca-sensitive TCA cycle enzymes, including pyruvate dehydrogenase, isocitrate dehydrogenase and α-ketoglutarate dehydrogenase, as well as Complexes I, III and V (ATP synthase) of the ETC, that are reported to be Ca-sensitive[21,22]. This would create bottlenecks at these various key enzymes. Decreased activity of ATP synthase would lead to a Na-induced decrease in ATP supply and an elevation in the proton motive force. Decreased activity of Complex III would lead to reduction of the CoQ pool to

$CoQH_2$, which, in conjunction with an elevated proton motive force will lead to reverse electron transport (RET) at Complex I. This therefore offers a mechanistic basis for the generation of ROS under conditions of elevated Na (Fig. 8b). Our proposed mechanism has similarities with a recent report that succinate accumulation due to ischaemia leads to reperfusion injury because of RET at Complex I, leading to a burst of ROS at the time of reperfusion[25]. In ischaemia/reperfusion (I/R), this burst of ROS could be prevented by inhibiting Complex II at the time of reperfusion with malonate ester pro-drugs. However, in our model, the generation of ROS by high Na precedes the

elevation of succinate and therefore it is mechanistically different to that observed during reperfusion injury.

In our experiments, Na elevation resulted in a significant decrease in ATP levels (Fig. 2a) but not PCr (Fig. 2b); the latter is usually deemed a more sensitive index of acute alterations in energetic demand which was maintained constant in our experiments. The reduction in ATP levels but not PCr during Na elevation was accompanied by a decrease in $\Delta G_{ATP}$ (Fig. 2d)[31]. CK activity for the conversion of PCr to ATP was unchanged while the rate of ATP synthesis from Pi and ADP was lower (Fig. 2g). These results are consistent with clinical observations where ATP synthesis is reduced in patients with HF[32]. Furthermore, the ATP levels remained diminished even after $[Na]_i$ was returned to baseline levels, despite restoration of the rate of ATP synthesis and $\Delta G_{ATP}$. The recovery of $\Delta G_{ATP}$ despite a sustained reduction in ATP levels is a result of the restoration of Pi levels and possible buffering of ATP to maintain PCr levels constant. Importantly, the observation that ATP levels were decreased while PCr levels were maintained suggests an impairment in ATP supply, thus implicating alterations in mitochondrial metabolism and metabolic output. These observations are consistent with the reported Ca-sensitivity of ATP synthase that would cause a reduction in ATP supply.

Mitochondrial metabolism was assessed using $^1H$ and $^{13}C$ NMR spectroscopy of metabolites labelled with [1-$^{13}C$] glucose during the Na elevation period or during its washout period. Na elevation resulted in increased TCA intermediates citrate, succinate, and fumarate (Fig. 3d–f) as well as anaplerotic substrates glutamate and aspartate (Fig. 3g, h), suggesting bottlenecks in TCA cycle flux due to inhibition of the key Ca-sensitive dehydrogenases. Consistent with the accumulation of TCA cycle metabolites, glycolytic metabolism was upregulated in Langendorff hearts under conditions of elevated Na (Fig. 3b, c) as an alternative pathway for ATP production. These metabolic changes were reversed on ouabain washout. Decreased activity of Complexes I, II and III of the ETC was also observed during Na elevation which also returned to baseline on ouabain washout. In addition, decreased respiration, as assessed by rate of oxygen consumption, was observed in isolated, intact cardiomyocytes treated with ouabain (Fig. 4A) and in mitochondria artificially exposed to rising concentrations of Na (Fig. 4b–d).

Succinate accumulation during Na elevation in our model is of interest due to the role that succinate plays in controlling reperfusion injury via RET[25]. In I/R, succinate accumulates during ischaemia due to hypoxia-driven impairment of ETC activity. Succinate is rapidly oxidised upon reperfusion, releasing electrons at a rate faster than can be consumed by Complex III. Thus, the excess electrons are forced back into Complex I in the reverse direction, producing ROS and causing ROS-induced injury during reperfusion. Interestingly, in our Na elevation model, ROS production begins concurrently with Na elevation (Fig. 6b) and *prior* to succinate accumulation, thus suggesting that the functional deficiency observed upon ouabain washout may be due to ROS-mediated myocardial injury occurring as early as during the Na elevation period and sustained throughout the washout period (Fig. 8a, b). In our model, the build-up of succinate can be explained through the measured reduction in Complex II activity or its reversal via RET. However, Complex II is not reported to be directly Ca-sensitive and therefore may be a secondary effect. We propose that it is the decreased activity of Complex III that is the primary driver– lower Complex III activity decreases the CoQ pool causing an increase in $CoQH_2$, therefore causing a subsequent decrease in Complex II activity (Fig. 8a). Indeed, we observed a decrease in the oxidised CoQ pool and a concomitant increase in the reduced $CoQH_2$ pool in Na elevation, which was reversed upon Na washout.

Taken together our findings can be explained through the activation of NCLX by elevated Na which causes a reduction in mitochondrial matrix Ca and its reversal on ouabain washout. Consistent with previous reports[1], in our hearts, preventing this depletion of

$[Ca]_m$, through targeted inhibition of NCLX with CGP-37157, prevented the metabolic and energetic alterations which were otherwise observed during Na elevation (Figs. 2 and 3). We further tested the hypothesis that the metabolic consequences of Na elevation are a result of the decreased CoQ pool using MitoQ, a mitochondria targeted antioxidant (Fig. 8c). We found that treatment with MitoQ improved the CoQ pool during Na elevation compared to sham (Fig. 6a, b). MitoQ treatment also prevented the energetic and metabolic alterations observed during Na elevation (Fig. 6f–k) in a similar manner to inhibition of NCLX. We therefore propose that the mechanism involves MitoQ acting as an electron acceptor in the absence of CoQ, which has been previously shown to be efficiently reduced by Complex II (but not Complexes I and III)[33]. The reduced $MitoQH_2$ is then re-oxidised to MitoQ when it scavenges and reacts with ROS under conditions of elevated Na and is available for re-reduction at Complex II (Fig. 8c).

The elevation of succinate and fumarate during Na elevation is of further interest since these metabolites have been previously implicated in mediating "metabolic signalling" in cells. Specifically, succinate and fumarate are inhibitors of the HIF-1α regulators PHDs and are thus able to induce HIF-1α stabilization independent of tissue hypoxia[26,27]. The time-dependent stabilization of HIF-1α in our hearts (Fig. 5a) under normoxic conditions (Fig. 5c, d) is consistent with this mechanism. The upregulation of HIF-1α during Na elevation may serve a protective function for the mitochondria. Previous studies by Li et al. show a direct association of HIF-1α with the mitochondria where HIF accumulation prevented the collapse of mitochondrial membrane potential and decreased mitochondrial apoptosis under conditions of oxidative stress[34]. However, as a transcription factor for many enzyme targets, the stabilization of HIF-1α also offers a mechanism by which chronic elevation of Na may lead to longer term metabolic remodelling seen in HF.

In conclusion, in this study, we have shown that acute changes in $[Na]_i$ alone, in the absence of disease, are sufficient to control mitochondrial function that leads to metabolic derangement, energetic deficiency and the generation of ROS. Alterations in metabolism were mediated through activation of NCLX that causes a decrease in $[Ca]_m$. Elevated Na led to a pseudo hypoxic response and stabilization of HIF-1α even under conditions of normal tissue oxygenation. Importantly, the metabolic alterations were reversible upon lowering Na back to baseline levels. Collectively, these results suggest that elevation of intracellular Na concentration is an early event in the metabolic dysfunction that drives the alterations in mitochondrial metabolism and myocardial energetics that underlie disease progression. It is notable that the hypo-phosphorylation of phospholemman that accompanies heart-failure-induced Na/K ATPase dysfunction can be measured as early as 3 days after a hypertrophic stimulus[35]. Our findings are of further relevance in diabetic cardiomyopathies, where Na is pathologically elevated as a consequence of insulin signalling and aberrant glucose transport and metabolism[36]. Therapeutic interventions aimed at preventing or reversing pathological Na elevation may therefore not only improve systolic and diastolic function in HF but may also be able to reverse adverse metabolic remodelling. Our work justifies a concerted search for agents that activate Na efflux by the Na/K pump for treatment of heart failure.

## Methods
### Approvals for animal work
Animal procedures were performed in compliance with Home Office Guidance on the Operation of the Animals (Scientific Procedures) Act of 1986, the Directive 2010/63/EU of the European Parliament, following a registered IACUC protocol (no. T11.2) and the King's College London institutional guidelines. Wistar rats (200–250 g) were purchased through Envigo. C57b/6N mice (3–4 months of age) were purchased through Janvier labs. All animals were housed according to

UK Home Office regulations at 20–24 °C, 45–65% humidity and 12 h/ 12 h light/dark cycle. Animals were killed humanely by anaesthetic overdose (intraperitoneal pentobarbital) and death confirmed by exsanguination.

## MitoQ treatment

Mitoquinone (MitoQ) was obtained from MitoQ Ltd (Auckland, New Zealand). Rats (150–170 g weight) were administered 500 μM MitoQ in drinking water for 2 weeks. Sham animals received normal drinking water.

## Langendorff perfusion

Rat hearts were rapidly excised and cannulated via the aorta and perfused at 37 °C at constant pressure of 80 mmHg. Hearts were perfused with a modified Krebs-Henseleit buffer (KHB) containing (in mM): 118 NaCl, 5.9 KCl, 1.16 MgSO$_4$, 2.2 CaCl$_2$, 25 NaHCO$_3$, 0.48 EDTA, 1 Na L-lactate, 0.1 Na pyruvate, 0.5 L-glutamic acid, 5 glucose, 0.4 intralipid and 5 mU/L insulin, and bubbled with 5% CO$_2$/95% O$_2$. For Na elevation experiments, hearts were subject to one of two perfusion protocols: (1) 20-min baseline followed by Na elevation (20 or 50 min) or (2) 20-min baseline followed by 20-min Na elevation and 20-min washout. Intracellular Na was elevated using 75 μM ouabain (Sigma) for 5 min to confirm positive inotropy. Then, 150 nM para-aminoblebbistatin (Cayman) was titrated into the heart to reverse the positive inotropy. For CGP experiments, hearts were perfused with 1.5 μM CGP-37157 (Insight Biotechnology) at the same time as ouabain; CGP was not used during washout. For $^{13}$C metabolomics in Na elevation, hearts were perfused in Krebs in which the glucose replaced with [1-$^{13}$C] glucose for subsequent metabolomic analysis by NMR. Hearts were perfused with $^{13}$C glucose during the last 10 min of the perfusion protocol before being snap-frozen in liquid nitrogen. Cardiac function was monitored throughout the protocol via a balloon inserted into the left ventricle and monitored on LabChart. The Langendorff rig was modified for NMR as described previously[35].

For Ca elevation experiments, rat hearts were subjected to 20-min baseline perfusion followed by 20-min Ca elevation. Hearts were perfused with the modified KHB containing 3.5 mM Ca for 5 min to stimulate a similar inotropy to that observed with Na elevation without altering intracellular Na concentration. Then, 150 nM PAB was titrated into the heart to reverse the Ca-driven inotropy. Hearts were snap-frozen in liquid nitrogen immediately after Ca elevation without washout.

## Cardiomyocyte isolation

Rat ventricular myocytes were used for metabolic flux assays and were isolated from Langendorff-perfused rat hearts as previously described[36]. Mouse ventricular cardiomyocytes were used for ROS assays and were isolated from adult, 10–16 weeks old C57BL/6N mouse hearts using collagenase type II (Worthington) in nominally Ca$^{2+}$ free perfusion buffer as previously described[37].

## Metabolite extraction

Rat hearts were immediately snap-frozen while the heart was still on the perfusion canula using Wollenberger tongs in liquid nitrogen and then stored at −80 °C prior to methanol (MeOH)/chloroform dual-phase extraction[38]. Snap frozen hearts were pulverised to fine powder over liquid nitrogen. Approximately 0.8–1 g of cardiac tissue was used for the extraction. The tissue was homogenized using a TissueRuptor in 4 mL of ice-cold methanol over ice. Subsequently, the following was added with vortexing between additions: 2 mL ice-cold ddH$_2$O, 2 mL ice-cold chloroform, 1.8 mL ice-cold ddH$_2$O and 2 mL ice-cold chloroform. Samples were centrifuged at 3600 × $g$, 4 °C for 60 min. The aqueous phase was transferred to a fresh tube containing Chelex and centrifuged for an additional 5 min. The supernatant was transferred to pre-weighed fresh tubes containing 15 μL pH indicator and snap frozen

in liquid nitrogen until frozen and then freeze-dried (for ~24 h at −100 °C and 0.01 mbar) with a hole made in the lid of each tube to allow for the sublimation to escape the tubes. Dried extracts were then stored at −80 °C prior to NMR analysis.

## High resolution $^1$H and $^{13}$C NMR spectroscopy

NMR spectra were acquired on a Bruker Avance III 400 MHz Spectrometer 9.4 T vertical-bore magnet using a 5 mm BBO probe. Samples (randomised and blinded) were transferred to 5 mL NMR tubes and subject to $^1$H 1D NOESY presat (noesygppr1d) acquired with 128 scans, experiment duration of 12 min 30 s, 32 k data points, a pre-scan delay of 3 s and acquisition time of 2.7 s. A line broadening factor of −0.3 Hz was applied prior to Gaussian multiplication, Fourier transformation, phase and baseline correction. Spectra were referenced to the TSP peak at 0 ppm and QA was performed to ensure good residual water suppression, flat baseline and optimal shimming (<1 Hz half peak width of the TSP peak). Assignment of metabolites to their respective peaks was carried out based on previously obtained data, confirmed by chemical shift using Chenomx and referenced to published data. Peak integration was performed in Topspin 3.7pl to create an intrng file that was then copied to all data and each spectrum visually inspected nad manually adjusted for any peak drift. Metabolite concentrations were calculated relative to the known concentration of the TSP peak. 2D gradient selected $^1$H/$^{13}$C HSQC (hsqcetgp) experiments were acquired with 16 scans, 1024 data points in the direct dimension, 512 data points in the indirect dimension and an experiment duration of 3 h 30 min. 2D HSQC data were processed with a QSINE function in both dimensions prior to Fourier transformation, phase, and 2D baseline correction. The same offset frequency taken from the TSP peak of the $^1$H 1D spectrum was applied in the $^1$H dimension of the 2D. Peak assignments were performed using the respective assignments from the 1D $^1$H data. Two-dimensional peak integration was performed in Topspin and normalised relative to the TSP peak.

## $^{23}$Na NMR spectroscopy

In situ $^{23}$Na NMR spectroscopy of perfused hearts was performed on a Bruker Avance III 400 MHz Spectrometer 9.4 T vertical-bore magnet using a 15 mm dual tuned $^1$H/$^{23}$Na microimaging coil. Spectra were acquired and analysed using TopSpin 3.7pl software. Shimming was carried out using a simple pulse-acquire on the $^{23}$Na channel (total Na signal). Time-resolved $^{23}$Na spectra were acquired using multiple quantum filtered (MQF) experiments to separate intracellular and extracellular contributions to the $^{23}$Na signals, as described previously[35]. Briefly, interleaved triple quantum filtered (TQF, 6 step phase cycle) and double quantum filtered (DQF, 4 step phase cycle) NMR acquisitions, consisting of 192 scans and an experimental duration of 1 min were recorded throughout the entire protocol with a pre-scan delay of 100 ms, 2 k data points and an acquisition time of 0.19 s. A line broadening factor of 15 Hz was applied prior to exponential multiplication, Fourier transformation, phase and baseline correction. Peak areas were measured in Topspin and normalised to the corresponding peak during the baseline stability period.

## $^{31}$P NMR spectroscopy

In situ $^{31}$P NMR spectra of perfused hearts were acquired on a Bruker Avance III 400 MHz Spectrometer 9.4 T vertical-bore magnet using 15 mm dual tuned $^1$H/$^{31}$P microimaging coil. Hearts were shimmed using a simple pulse-acquire on the $^1$H channel (H$_2$O resonance) to yield a water linewidth of <50 Hz. $^{31}$P spectra were acquired with a 60° flip angle, 64 scans and a total experiment duration of 4 min employing a pre-scan delay of 3 s, 16 k data points and an acquisition time of 0.85 s. A line broadening factor of 25 Hz was applied prior to exponential multiplication, Fourier transformation, phase, and baseline correction. Peak areas of the Pi, PCr and β-ATP peaks were measured in Topspin and normalised to the corresponding peak during the

baseline stability period. The chemical shift difference between the PCr and Pi peaks (Δppm) was measured and intracellular pH and $\Delta G_{ATP}$ were calculated as previously described[38], using creatine concentrations obtained from $^1$H spectra of each heart (mean value 15.7 mM). $^{31}$P saturation transfer experiments were acquired using a standard pre-saturation sequence (zgpr) and shifting the radio frequency pulse either to midway between the γ and α-ATP resonances (control spectrum), or on resonance with the γ-ATP peak. The resulting peak integrals of the unsaturated PCr and Pi peaks were subtracted from the same peak integrals in the reference spectrum to give the magnetization transfer $\Delta M_z$ that occurred during the 3 s pre-saturation period.

### $^{64}$Cu·CTS radiolabelling

$^{64}$Cu was produced and used to radiolabel 2,3-pentanedione bis(thiosemicarbazone) (CTS) as previously described[39]. Rat hearts were Langendorff perfused at a constant flow of 14 mL/min with KHB buffer and gassed with 95% $O_2$/5% $CO_2$ mixture, and contractile function was measured with a left ventricular balloon. Radiotracer uptake and pharmacokinetics were monitored as previously described[39]. Briefly, a 100 μL bolus of radiotracer (1 MBq) was injected into the arterial line at the following time-points: 10 min, 30 min and 60 min, corresponding to end of baseline, 20 min ouabain and 50 min ouabain time-points, respectively. Tissue retention was calculated as the residual activity in the heart 10 min after injection as a percentage of the peak activity (percentage injected dose [%ID]). Time-matched controls received radiotracers at the corresponding time-points without ouabain treatment; hypoxic hearts were perfused in anoxic buffer (0% $O_2$/5% $CO_2$) as positive control for tissue hypoxia.

### Metabolic flux assays

Mitochondrial oxygen consumption was measured using the mito stress test on the Seahorse XF24 system in intact or permeabilised myocytes. Isolated rat ventricular cardiomyocytes were plated at 12,000 cells/well onto laminin-coated, Seahorse XF24 cell culture microplate and allowed to settle for 30 min at 37 °C. For intact myocytes, cell medium was exchanged to the Seahorse base medium supplemented with 5 mM glucose and 1 mM sodium pyruvate (supplemented medium), pH 7.4 at 37 °C and cells were allowed to acclimatise for 1 h. Cells were then pre-treated with 10 μM para-amino blebbistatin in supplemented medium for 20 min prior to measurements. Injection port A was loaded with 0.5 or 1 mM ouabain for a final concentration of 50 or 100 μM ouabain. Control wells received equal volume of supplemented medium; some wells also carried 10 μM CGP-37157 alongside ouabain in port A for a final concentration of 1 μM. Following a 12 min equilibration period, 4 measurements were taken at each condition with 30 s mixing, 30 s waiting, and 2 min measurement per cycle. For permeabilised myocytes, cells were washed twice in Ca-free Tyrode buffer and twice in mitochondrial respiration buffer (MRB) containing (in mM): 0.5 EGTA, 3 MgCl2, 20 taurine, 10 K2HPO4, 20 HEPES, 0.1% fatty acid free BSA, 60 potassium lactobionate, 110 mannitol, 0.3 dithiothreitol (DTT), pH 7.4 at 37 °C. Cells were permeabilised using 50 μg/mL saponin immediately prior to loading plates into the plate reader. Injection port A was loaded with 50, 100 or 200 mM NaCl in MRB; some wells carried 10 μM CGP-37157 alongside 200 mM NaCl. Control wells received equal volume of MRB. Injection port B and C were loaded with 50 mM malate/100 mM Na pyruvate, pH 7.4 at 37 °C and 10 mM ADP, pH 7.4 at 37 °C, respectively. Each port was loaded with 10X of the final concentration of the respective treatments. For each condition, 4 measurements were taken with 30 s mixing, 30 s waiting, and 2 min measurement per cycle.

### ETC complex assays

Rat cardiac homogenates were prepared using approximately 10 mg of frozen, crushed tissues suspended in 200 μL of ice-cold KME buffer (in mM): 100.0 KCl, 50.0 Mops, 0.5 EGTA (pH 7.4 at room temperature

with NaOH). Samples were homogenised by rupturing with a TissueRuptor (Qiagen) over ice and used immediately to measure the activities of mitochondrial complexes.

Complex I activity was measured as the rate of NADH oxidation at 340 nm and 30 °C using a spectrophotometer. The assay buffer contained the following (in mM, unless otherwise specified): 25 potassium phosphate (pH 7.2 at room temperature), 5 MgCl2, 0.13 NADH (MP Biomedicals, LLP), 3.65 antimycin A (Santa Cruz), 65 μM coenzyme Q1 (Sigma), and 250 mg fatty acid free BSA (Sigma). The reaction was carried out in 1 mL assay buffer supplemented with 20 μL of protein lysate and read for 1 min at 340 nm against a blank containing ddH2O. As a negative control, 200 μM of rotenone (Sigma) was added to a new reaction and the inhibited rate measured for 1 min. The activity of Complex I was determined by dividing the gradient of the absorbance change over the extinction coefficient (6810 nmol·min$^{-1}$) and expressed in nmol·min$^{-1}$·μg protein$^{-1}$.

Complex II activity was measured as the rate of succinate-dependent reduction of dichlorophenolindophenol (DCPIP) at 600 nm and 30 °C using a spectrophotometer. The assay buffer contained the following (in mM): 25.0 potassium phosphate pH 7.2, 5.0 MgCl2, and 2.0 sodium succinate dibasic hexahydrate (Sigma-Aldrich). The reaction was carried out in 985 μL of assay buffer supplemented by 10 μM L of buffer containing 365 μM antimycin, 50 μM DCPIP (Sigma-Aldrich) and 500 μM rotenone, followed by addition of 65 μM CoQ1 and 20 μL protein lysate and read for 2 min at 600 nm against a blank containing ddH2O. As a negative control, 200 μM of sodium malonate (Sigma-Aldrich) was added to a new reaction and the inhibited rate measured for 2 min. The activity of complex II was determined by dividing the gradient of the absorbance change over the extinction coefficient (21,000 nmol·min$^{-1}$) and expressed in nmol·min$^{-1}$·μg protein$^{-1}$.

Complex III activity was measured as the rate of reduction of cytochrome $c^{3+}$ at 550 nm and 30 °C using a spectrophotometer and reduced ubiquinol as an electron acceptor. Preparation of reduced ubiquinol. Ubiquinol was prepared as follows: 10 mg of decylubiquinone (Sigma-Aldrich) was dissolved in 312.5 μL absolute ethanol to give a final concentration of 100 mM. An aliquot of 100 μL decylubiquinone working solution was further diluted in 900 μL ethanol and acidified to pH 2 with a 6 M HCl solution. The pH was checked with a pH indicator paper. The ubiquinone was then reduced with a pinch of sodium borohydrate (Sigma-Aldrich) and 1 mL of ddH2O was added to stop the reaction. Excess sodium borohydrate was allowed to settle and the sample centrifuged briefly to separate the sodium borohydrate precipitate from the reduced ubiquinol. Using a pH indicator paper, the pH of the ubiquinol was verified to be at pH 2 before using in the activity assay. Activity assay: The assay buffer contained the following (in mM, unless otherwise specified): 50.0 potassium phosphate pH 7.2, 3.0 sodium azide (Sigma-Aldrich), 1.5 μM rotenone, and 50 μM cytochrome c from bovine heart (Sigma-Aldrich). The reaction was carried in 1 mL of assay buffer, supplemented with 5 μL reduced ubiquinol and 20 μL protein lysate and read for 1 min at 550 nm against a blank containing ddH2O. As a negative control, 200 μM of antimycin A was added to a new reaction and the inhibited rate measured for 1 min. The activity of complex III was determined by dividing the gradient of the absorbance change over the extinction coefficient (19,100 nmol·min$^{-1}$) and expressed in nmol·min$^{-1}$·μg protein$^{-1}$.

### CoQ pool mass spectrometry

Frozen pulverised rat cardiac tissue was homogenised with a Precellys tissue homogeniser in an extraction buffer which contain 250 μL methanol with 0.1% HCl and 250 μL hexane. The two-layer supernatant was transferred to a new tube and centrifuged at 17,000 × $g$ for 5 min at 4 °C. The upper layer was transferred to a mass spectrometry vial and blown with $N_2$ gas at 37 °C. To the sample, 250 μL methanol and 2 mM Ammonium Formate was added and the sample was analysed by the

Waters Mass spectrometry TQ-S and the software Masslynx. Data was analysed in terms of the ratio of % of Reduced $CoQH_2$ and the total CoQ (sum of reduced $CoQH_2$ and oxidized CoQ) for $CoQ_9$ and $CoQ_{10}$.

## ROS measurement

Adult mouse ventricular myocytes were stained with 5 μM MitoSOX mitochondrial superoxide indicator (M36008, Thermo fisher) in Tyrode buffer containing (in mM): 140 NaCl, 5.4 KCl, 2 $MgCl_2$, 1.8 $CaCl_2$, 10 HEPES, and 10 glucose (pH 7.4 at 37 °C) at 37 °C for 15 min and washed twice buffer before imaging. Cells were excited at 550 nm and the emitted light was recorded at 570 nm using the IX83 microscope and the MT20 illumination system (Olympus). Light intensity was set to 4% and the exposure time was 500 ms. Each image was recorded at an interval of 120 s. After the fluorescence intensity stabilised as baseline readings, DMSO or ouabain at different concentrations was added, respectively. Cells were then washed with Tyrode buffer to remove ouabain. The Tyrode solution used for imaging contained 10 μM Blebbistatin. After the measurement, fluorescent data were extracted by the cellSens software. The fluorescent intensities of each cell at different time points were normalised against the mean baseline of its own. The measurement was repeated three times on different days with different isolations of ventricular cardiomyocytes from mice with 3–4 months of age independently.

## Western blot

Protein lysate was made from frozen, crushed tissues in RIPA buffer supplemented with 1:100 protease inhibitor cocktail (Roche) and briefly sonicated to maximize tissue rupture. For SDS-PAGE, 100 μg protein lysate was loaded into a 10% gel and transferred to a 0.45 μM PVDF membrane on the Transblot Turbo transfer system. Blots were blocked overnight at 4 °C with primary antibody and 1 h at room temperature with HRP-conjugated secondary antibody. Blots were exposed on film. Band densities were quantified in ImageJ. Antibodies used were rabbit polyclonal anti-HIF-1α (Novus, NB100-479) and rabbit monoclonal anti-GAPDH (Abcam, EPR16891 goat anti-rabbit-HRP (Santa Cruz, sc-2030).

## Statistics

All data are reported as mean ± SEM unless otherwise specified. Either a two-tail, unpaired student's $t$-test or two-way ANOVA was used to compare the means of different treatment groups. *$P < 0.05$, **$P < 0.01$, ***$P < 0.001$, ****$P < 0.0001$. Mitochondrial oxygen consumption data was analysed using a nested (hierarchical) one-way ANOVA. Data analysis was carried out in a blinded manner. Number of replicates are stated in each figure legend. For experiments on single cells, repeats are noted as $n$ = number of animals or $n$ = number of cells from $N$ = number of animals from which cells were isolated. Significant $P$ values are indicated by *$P < 0.05$, **$P < 0.01$, ***$P < 0.001$, ****$P < 0.0001$.

## Reporting summary

Further information on research design is available in the Nature Portfolio Reporting Summary linked to this article.

# Data availability

Source data, including all metabolomic data, are provided as a Source Data file. All other unprocessed data supporting the findings of this study are available from the corresponding authors upon reasonable request. Source data are provided with this paper.

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

## Acknowledgements
This work was supported by British Heart Foundation Programme Grant RG/12/4/29426 and RG/17/15/33106 (M.J.S. and W.F.); NIHR Biomedical Research Centre at Guy's and St Thomas' NHS Foundation Trust and KCL (T.R.E. and R.S.); EPSRC Programme grants "MITHRAS" EP/S032789/1 and "RedOx-KCL" EP/S019901/1 (R.S. and F.B.); German Research Foundation GRK2824 and KA1269/13-1 (J.G. and D.M.K.); Medical Research Council UK (MC_UU_00028/4); Wellcome Trust Investigator award 220257/Z/20/Z (M.P.M.); the Centre of Excellence in Medical Engineering funded by the Wellcome Trust and Engineering and EPSRC WT 203148/Z/16/Z (T.R.E. and R.S.) and the BHF Centre of Research Excellence RE/18/2/34213. We thank Dr Mufassra Mushtaq (University Medical Centre Göttingen, Germany) for technical assistance in myocyte preparation for the MitoSox experiments.

## Author contributions
Y.J.C., Z.H., F.B., C.S.Y. and J.G. completed all the experimental work described here. Y.J.C., M.J.S., T.R.E., W.F., R.S., M.P.M. and D.M.K. were responsible for hypothesis generation and experimental design. The work was supported by grants to M.J.S., W.F., T.R.E., R.S., D.M.K. and M.P.M. The original manuscript draught was written by Y.J.C., M.J.S. and T.R.E. and all authors contributed to revision and editing.

## Competing interests
The MitoQ was an unconditional gift from MitoQ Inc, Auckland, New Zealand. M.P.M. is on the Scientific Advisory Board of MitoQ Inc. and holds stock in the company. All other authors declare no competing interests.
