## [Peer Review File · Nature Communications]

REVIEWER COMMENTS

Reviewer #1 (Remarks to the Author):

The limited novelty of current manuscript is one of my main concerns. The same group of authors published a similar study [Nature communication 2020 (11)4337] and a review (JMCC2021 August 08; 161: 106–115. doi:10.1016/j.yjmcc.2021.08.003]. The current version provides neither conceptual novelty nor extensions of its therapeutic advancements. However, I do believe the impacts disease relevant of proposed mechanism are very high. The authors are strongly encouraged to include the experiments showing whether lowering intracellular sodium can delay or prevent the onset of heart diseases in an established model. The new data showing lowering intracellular sodium can improve the cardiac metabolism and cardiac function in an established model of heart failure is highly desired. The inclusion of this data would make this manuscript more impactful and proposed mechanism more convincing.

Reviewer #2 (Remarks to the Author):

This is a very nice follow-up study to a 2020 Nat Comm paper from the same group. There is much new data and insights here and the paper is well-written and also fills in key mechanistic details as a nice comprehensive study. While one might say that it provides incremental conceptual advancement on their prior paper identifying a novel high $[Na]_i$ – induced acute modulation of cardiac myocyte metabolism. Nevertheless, I think this new work is a very good paper and of great interest to the cardiovascular field.

They use acute Na-pump inhibition (ouabain) to elevate $[Na]_i$ in cardiac myocytes, while preventing increased cardiac work by cleverly titrating down LVDP by inhibiting myosin crossbridge function (and energy consumption). In their 2020 paper they showed that this acute elevation of myocyte $[Na]_i$; altered mitochondrial energy metabolism (substrate utilization and TCA cycle intermediates). Moreover, this mimicked chronic changes they observed in acquired cardiac disease (TAC-induced hypertrophy) and genetic alteration of phospholemman to limit Na-pump function. That opened up a new acute metabolic rewiring framework that they delve more deeply into here.

The new paper shows “reversibility” by removing the ouabain and showing that the effects are reversed over the course of only tens of minutes and that this is comparable to the prevention by NCLX inhibition by CGP. They also add interesting new data on reduced ATP levels, lower activity of

Complex I, II and III in mitochondrial electron transport chain and increased ROS production and recruitment of HIF-1alpha (in the absence of hypoxia). These are all substantial additions which allow them to better describe and mechanistically connect the involved pathways. I do have some additional specific comments.

1. They emphasize (in many places) that these metabolic/energetic consequences are a direct effect of elevated $[Na]_i$, but the effect is clearly mediated indirectly by $[Na]_i$, because the high $[Na]_i$ causes a reduction of intra-mitochondrial $[Ca]$ via the mitochondrial Na/Ca exchange mNCX. Indeed, the inhibition of mNCX with CGP prevents the alterations in metabolism induced by high $[Na]_i$. The mitochondrial targets implicated are indeed, mainly those that are known to be Ca-sensitive. I suggest modifying the wording (at least in using “direct” in that context).

2. Line 118. Starting here and in Fig 2 on, I assume all intact heart experiments are using the combination of 75 μ M ouabain + 150 nM PAB. That is often left unstated and even unclear in figures and legends. I only realized later that the PAB titration probably occurs for each heart and that is why there is a transient increase as ouabain is applied.

I think I understand the codes for bars in Fig 2A-D, but maybe these would be clearer as follows:

Delete the arrow Na and w/o from the time controls and put the min at which measure taken instead.

Use two lines for each pair of bars Time Control for first pair (top & bottom), Ouab +PAB for second pair and CGP Ouab +PAB (if the CGP was pretreatment). The arrow and Na in the time control is also confusing, maybe you could put the time in min when the $[Na]$ and w/o data are taken in the other pairs (since no solution changes occurred in the tmc).

3. Line 133-148. The reduced ATP with normal PCr is puzzling. Is there evidence that the total AMP-ADP-ATP pool size is smaller which would allow CK reaction to still be in balance?

4. Lines 177, 187 - a couple of examples where “direct effect of cytosolic $[Na]$ ” annoyed me. It bugs me (I think) because it’s really the mitochondrial $[Ca]$ which is directly mediating the effects on numerous intramitochondrial proteins (e.g. dehydrogenases, redox enzymes). Of course the $[Na]_i$ is modulating that mitochondrial $[Ca]$ (which is not measured anywhere here), so my issue is mainly the word direct. Along these lines, many of the effects of Na reported are already known Ca targets, so while I like the work your line 210 are more on-point on how Na indirectly modulates mitochondrial activity.

5. Line 252: In Fig 6C-D the 0.1 mM – 10 mM refer to ouabain concentrations? Be clearer. In panels 6A and B was the ouabain with PAB

6. Around here I got confused about which experiments were in rats and which in mice. Making the species appear in the Figure title would help keep this from being ambiguous.

7. Line 274: add comma after HIF-1a to sharpen intent.

8. Line 285 & 297: I think you mean Fig 7C-D and Fig 8, respectively.

9. Line 346, 347 (and elsewhere) washout is really of ouabain and not Na, although $[Na]_i$ levels will decay as ouabain washes out.

10. Line 397-8: This seems a bit overstated because it is unknown how early $[Na]_i$ rises during the development of heart failure.

Reviewer #3 (Remarks to the Author):

This is a potentially interesting paper investigating the role of Na on mitochondrial metabolism of the heart. As suggested by the editor, the authors have indeed already studied this link in the past. However, I agree with the authors that the work presented here is novel, as they are now trying to provide more detailed insight into its mechanism. And I say “trying to” since some of their conclusions are based on ^{31}P NMR experiments that are likely the weakest point of the paper.

The findings of dropping ATP are in direct contrast to the authors' previous work showing stable PCr/ATP under increased Na conditions. These changes are unexpected also in relation to literature where PCr rather than ATP levels drop (or both in case of severe heart failure), while the PCr levels stay constant in this work. The authors try to address this issue with performing saturation transfer experiments, but these are, unfortunately, not done correctly, and the conclusions drawn are based on wrong assumptions.

Main issues:

- 1) saturation of gamma-ATP allows for calculation of the forward rate constant, forward as meaning ATP synthesis, i.e. PCr-to-ATP, not the other way around as you suggest; To explain, the drop in PCr signal is caused by the ^{31}P nuclei leaving PCr and being replenished by saturated nuclei (no signal), so it is the rate of ^{31}P nuclei leaving PCr molecule you are measuring; Hence, your results actually suggest stable ATP production via CK, undermining your decreased ATP supply argument.
- 2) I'm surprised you haven't investigated the Pi-to-ATP reaction, you have all data for it and this is potentially much more exciting for your study.
- 3) saturation of PCr alone is not sufficient to explore ATP hydrolysis rates since there is always the $\text{ATP} \leftrightarrow \text{Pi} + \text{ADP}$ reaction, and only through simultaneous saturation of both PCr and Pi the ATP hydrolysis can be estimated;
- 4) it is unclear what is the "control" saturation experiment supposed to be controlling, I can presume it is supposed to work as a control for the pointless PCr saturation experiment? Definitely can't account for any direct saturation for the PCr-to-ATP experiment where gamma-ATP is saturated. You might not need a control experiment at all, if your pulses are selective enough, but if you do, you need to place them mirrored around the exchange partner, i.e. in this case PCr;
- 5) Can you assume stable apparent T1 across the different hearts during Na infusion and wash-out? Since this has a significant effect on the exchange rate/flux, this would need to be justified;
- 6) In relation to your static spectra, I'm not sure you can call your data fully relaxed if cardiac PCr in mice at 9.4T is $2.54 \text{ s} \pm 0.41 \text{ s}$ (10.1002/nbm.3371) and Pi is likely longer. This would suggest you can't directly use your signals for the ΔGATP calculation as described in the referenced paper. You would need to apply a saturation correction first. In addition, the assumption of 20 mM free creatine seems too low (doi: 10.1016/0167-4889(92)90057-i) and since you are not actually calculating concentrations of your ^{31}P metabolites, it is difficult to put into perspective with your data.
- 7) While beta-ATP is indeed technically the best to use for ATP quantification, but given its resonance frequency (dependent on Mg concentration and pH and requiring large excitation bandwidth) and the fact it is a triplet, it might be less reliable to fit than the other two peaks. I would suggest double checking the unexpected results by analysing gamma ATP and/or an average of all three peaks.

Reviewer #4 (Remarks to the Author):

The effect of Na on the Ca-mediated regulation of mitochondrial bioenergetics is already known. In this study, however, the authors do provide a broader view of the cell wide consequences related to the effect of high sodium on cardiomyocyte bioenergetics in general. This was achieved by branching out from its effects on mitochondria and demonstrating how the Na-mediated depletion of mitochondrial calcium may cause "Metabolic Gridlock" characterized by decreased OXPHOS and increased ROS production, which leads to the activation of glycolysis, potentially through HIF-1 stabilization. However, this broader view is limited given the authors did not fully interrogate the

metabolic/cell wide consequences of this pathological mechanisms. It would be useful to interrogate these pathways further using cell lines treated with the Na/Ca exchanger inhibitor CGP or a CRISPR knockout. This would allow an in-depth assessment of the metabolic changes that occur to glycolysis and in mitochondria. Additionally, such a model would allow for the interrogation of mitochondrial ROS generation and the amount formed during RET (assuming that RET is activated in this model, which is in consideration of the fact that mitochondria contain 12 ROS generators). An assessment of HIF-1 activation would also be allowed through transcriptome analyses. Other transcription factors can also measured like NRF2, HSF-1, and NFkB.

Thus, I believe this study has a lot of merit and could be suitable for Nature Communications. But it falls short by missing the “big picture” in terms of the metabolic adaptations that are invoked in response to the Na overload and the pathological consequences that follow. Many more experiments would be required.

Responses to Editor and Reviewers' comments.

We thank the editors and the reviewers for their swift and constructive review process. We have now revised the manuscript addressing all the issues raised. A detailed itemised response is provided below. Each original comment is provided in *red italics* followed by our response. Corrections in the manuscript itself are highlighted in **yellow**.

Reviewer #1:

- 1. The limited novelty of current manuscript is one of my main concerns. The same group of authors published a similar study [Nature communication 2020 (11)4337]. The current version provides neither conceptual novelty nor extensions of its therapeutic advancements. However, I do believe the impacts disease relevant of proposed mechanism are very high.*

We thank the reviewer for their comment that the impact of the proposed mechanism in disease being very high. However, we respectfully disagree that it has limited novelty with respect to our previous *Nat Comms* paper. We note that Reviewer 2 found “*much new data and insights here and...fills in key mechanistic details as a nice comprehensive study*” and Reviewer 3 stated that “*the work presented here is novel...to provide more detailed insight into its mechanism*”. In our previous work we observed a switch in substrate preference under conditions of elevated Na but did not explore in detail the mechanisms by which Na impacts mitochondrial metabolism which we do here. We also didn't previously show the reversibility of such Na driven metabolic changes and therefore for the first time our work shows that reversing Na overload may be a tractable therapeutic target to reverse metabolic dysfunction if it could be achieved. With respect to this latter point, see below.

- 2. The authors are strongly encouraged to include the experiments showing whether lowering intracellular sodium can delay or prevent the onset of heart diseases in an established model. The new data showing lowering intracellular sodium can improve the cardiac metabolism and cardiac function in an established model of heart failure is highly desired. The inclusion of this data would make this manuscript more impactful and proposed mechanism more convincing.*

We completely agree - this is indeed to overall aim of our programme. Several studies have shown that ranolazine (which blocks Na influx), for example, can ameliorate heart failure symptoms. However, no studies to date have shown that ranolazine lowers intracellular Na in an established model of heart failure. Other agents such as NHE inhibitors, SGLT2i's, beta blockers etc have also been shown to ameliorate heart failure and an element of their protection has been suggested by some to involve Na lowering (i.e. Pilippaert: DOI: 10.1161/CIRCULATIONAHA.121.053350, Zuurbier: DOI: 10.1093/cvr/cvab129). However, many interventions such as these may achieve Na lowering as a secondary effect to ameliorating the disease through other mechanisms. It is, therefore, hard to ascribe their benefits to Na lowering alone. Such interventions will not prove a central role for Na if they do not exclusively target and lower Na as their only effect. We (Shattock: doi: 10.1016/j.coph.2008.12.015), and others (Rasmussen: PMID: 18052902), have proposed Na/K ATPase stimulation as a

proof-of-concept therapeutic intervention. However, to date, no agents have been identified that can achieve this.

This problem is doubly difficult when heart failure has already been established. The reviewer requests experiments to show benefit in an “*established model of heart failure*” and requests we show that lowering Na (when presumably already elevated) is beneficial. To our knowledge there are no current pharmacological agents that can achieve this. In fact, a main driver underpinning these studies is to incentivise a search for such therapies. In this regard we have proposed two inducible/activatable transgenic mouse models where we hope that we will be able to address this question directly in the future but at present neither mouse has been generated and we are awaiting funding for these studies. So, to summarise, we cannot currently think of a way to do these experiments (unless the reviewer can suggest a feasible Na-lowering approach?). Additionally, we hope that the current study will allow us to demonstrate to our funders that the metabolic changes that accompany heart failure are, at least in part, driven by Na accumulation and importantly are reversible. It is indeed a future aim to do this in real models of established heart failure but currently we do not have the tools. As such this is way beyond the scope of our current study which we argue is necessary to lay the foundation for such future work.

Reviewer #2:

This is a very nice follow-up study to a 2020 Nat Comm paper from the same group. There is much new data and insights here and the paper is well-written and also fills in key mechanistic details as a nice comprehensive study. While one might say that it provides incremental conceptual advancement on their prior paper identifying a novel high [Na]ⁱ – induced acute modulation of cardiac myocyte metabolism. Nevertheless, I think this new work is a very good paper and of great interest to the cardiovascular field.

We thank the reviewer for their positive remarks. We can agree that the conceptual advance is incremental in the sense that we follow on our previous work to better understand to impact of elevated Na on myocardial metabolism. However, we are happy the reviewer appreciated the new data and many new insights, not least two very important advances over our previous work, being the reversibility of acute Na induced metabolic dysfunction, and perhaps of greater impact we here provide a detailed description of the overall mechanisms at play, which were not detailed in our prior work.

They use acute Na-pump inhibition (ouabain) to elevate [Na]ⁱ in cardiac myocytes, while preventing increased cardiac work by cleverly titrating down LVDP by inhibiting myosin crossbridge function (and energy consumption). In their 2020 paper they showed that this acute elevation of myocyte [Na]ⁱ; altered mitochondrial energy metabolism (substrate utilization and TCA cycle intermediates). Moreover, this mimicked chronic changes they observed in acquired cardiac disease (TAC-induced hypertrophy) and genetic alteration of phospholemman to limit Na-pump function. That opened up a new acute metabolic rewiring framework that they delve more deeply into here.

Thank you for the supportive comments.

The new paper shows “reversibility” by removing the ouabain and showing that the effects are reversed over the course of only tens of minutes and that this is comparable to the prevention by NCLX inhibition by CGP. They also add interesting new data on reduced ATP levels, lower activity of Complex I, II and III in mitochondrial electron transport chain and increased ROS production and recruitment of HIF-1alpha (in the absence of hypoxia). These are all substantial additions which allow them to better describe and mechanistically connect the involved pathways. I do have some additional specific comments.

We are happy that the reviewer finds these to be “substantial” additions in support of our prior work.

- 1. They emphasize (in many places) that these metabolic/energetic consequences are a direct effect of elevated [Na]ⁱ, but the effect is clearly mediated indirectly by [Na]ⁱ, because the high [Na]ⁱ causes a reduction of intra-mitochondrial [Ca] via the mitochondrial Na/Ca exchange mNCX. Indeed, the inhibition of mNCX with CGP prevents the alterations in metabolism induced by high [Na]ⁱ. The mitochondrial targets implicated are indeed, mainly those that are known to be Ca-sensitive. I suggest modifying the wording (at least in using “direct” in that context).*

We apologise for our imprecise use of language. We completely agree. We have reworded all instances where we have used this terminology to clarify that these are indirect effects driven by Na lowering matrix Ca.

3. *Line 118. Starting here and in Fig 2 on, I assume all intact heart experiments are using the combination of 75 μ M ouabain + 150 nM PAB. That is often left unstated and even unclear in figures and legends. I only realized later that the PAB titration probably occurs for each heart and that is why there is a transient increase as ouabain is applied.*

Apologies. The description of the PAB titration in the Methods section (see line 431-433) and Figure 1 legend (see line 751-752) clarifies the timings of ouabain and subsequent PAB treatment in hearts. Briefly, PAB-free (+ouabain) buffer and buffer containing 150 nM PAB (+ouabain) were included in two parallel perfusion lines which converged at a 3-way tap proximal to our peristaltic perfusion pump and the aortic 'umbilical' line to the heart within the NMR magnet (see Eykyn et al – DOI: 10.1016/j.yjmcc.2015.07.009). As the ouabain-induced inotropy started to develop, this tap was intermittently and partially opened to allow mixing of the PAB-containing and PAB-free buffers – in this way titrating the inotropy back to the control level – hence the transient inotropy.

- (a) *I think I understand the codes for bars in Fig 2A-D, but maybe these would be clearer as follows:*

Delete the arrow Na and w/o from the time controls and put the min at which measure taken instead.

We have modified the figures as suggested by the reviewer.

- (b) *Use two lines for each pair of bars Time Control for first pair (top & bottom), Ouab +PAB for second pair and CGP Ouab +PAB (if the CGP was pretreatment). The arrow and Na in the time control is also confusing, maybe you could put the time in min when the Na and w/o data are taken in the other pairs (since no solution changes occurred in the tmc).*

We have modified the figures as suggested by the reviewer.

3. *Line 133-148. The reduced ATP with normal PCr is puzzling. Is there evidence that the total AMP-ADP-ATP pool size is smaller which would allow CK reaction to still be in balance?*

Thank you for this suggestion. We have revisited our ¹H NMR data acquired on extracted tissue snap frozen at the end of the respective ³¹P experiments. We have estimated the total adenine nucleotide pool (ATP+ADP+AMP) and find that it is not significantly different at 40 mins during Na elevation compared to its time matched control. The TAN pool is also not significantly different at 60 mins after ouabain washout with respect to its time matched control. We have included a statement to clarify this in the results section (see lines 135-137).

For clarity the data is reproduced here:

However, we note a trend towards lower TAN at 40 mins during elevated Na compared to baseline which is further depressed at 60 min washout. While the time matched control is unchanged at 40 mins, and maybe marginally lower at 60 mins, this is not statistically significant. Given the variability of this data we are not able to rule out the possibility that the TAN pool is decreased by elevated Na and this could offer a plausible explanation why we observed decreased ATP with unchanged PCr.

4. *Lines 177, 187 - a couple of examples where “direct effect of cytosolic [Na]” annoyed me. It bugs me (I think) because it’s really the mitochondrial [Ca] which is directly mediating the effects on numerous intramitochondrial proteins (e.g. dehydrogenases, redox enzymes). Of course the [Na]_i is modulating that mitochondrial [Ca] (which is not measured anywhere here), so my issue is mainly the word direct. Along these lines, many of the effects of Na reported are already known Ca targets, so while I like the work your line 210 are more on-point on how Na indirectly modulates mitochondrial activity.*

Agreed. As above, we apologise for our imprecise use of language. We have removed all instances referring to “direct” effects of Na.

5. *Line 252: In Fig 6C-D the 0.1 mM – 10 mM refer to ouabain concentrations? Be clearer. In panels 6A and B was the ouabain with PAB*

Yes, this has now been clarified in Figure 6 legend (see lines 816-817).

6. *Around here I got confused about which experiments were in rats and which in mice. Making the species appear in the Figure title would help keep this from being ambiguous.*

All experiments were carried out in rats with the exception of the ROS measurements in Figure 6C-D, which were carried out in isolated mouse myocytes. The species is specified in each figure legend when describing n numbers used for each experiment

(i.e. n=6 rats per group). The Methods section has been revised to further clarify the species used in each experiment (see lines 425, 441, 447-448, 452, 515, 536, 581, 589).

7. Line 274: add comma after HIF-1a to sharpen intent.

Done (new line 280).

8. Line 285 & 297: I think you mean Fig 7C-D and Fig 8, respectively.

Done (new line 288, 301).

9. Line 346, 347 (and elsewhere) washout is really of ouabain and not Na, although [Na]_i levels will decay as ouabain washes out.

All instances of *Na washout* have now been replaced by *ouabain washout* (see lines 245, 350, 351, 361, 372, 866).

10. Line 397-8: This seems a bit overstated because it is unknown how early [Na]_i rises during the development of heart failure.

Apologies, we completely agree, the early time course of Na elevation has not been delineated. However, it is notable that the hypo-phosphorylation of phospholemman that accompanies heart-failure-induced Na/K ATPase dysfunction can be measured as early as 3 days after a hypertrophic stimulus. We have therefore modified this statement as requested (see lines 401-410).

Reviewer #3:

This is a potentially interesting paper investigating the role of Na on mitochondrial metabolism of the heart. As suggested by the editor, the authors have indeed already studied this link in the past. However, I agree with the authors that the work presented here is novel, as they are now trying to provide more detailed insight into its mechanism. And I say “trying to” since some of their conclusions are based on ^{31}P NMR experiments that are likely the weakest point of the paper.

We are grateful for the reviewer's valuable comments and are pleased to note their confirmation that the work is indeed novel.

The findings of dropping ATP are in direct contrast to the authors' previous work showing stable PCr/ATP under increased Na conditions. These changes are unexpected also in relation to literature where PCr rather than ATP levels drop (or both in case of severe heart failure), while the PCr levels stay constant in this work.

We respectfully suggest that our findings are not in “direct contrast” to our previous work. In the Aksentijevic et al (2020) study the analysis of extracted metabolites, showed ATP+ADP slightly, but not significantly down and PCr slightly, but not significantly up (Figure 4b). This is not dissimilar to our observations in our present study where ATP is down and PCr is preserved. We do agree that when measured using ^{31}P -NMR in isolated hearts (Aksentijevic et al: Figure 4c - red bar) we showed no significant change in PCr/ATP ratio (red vs white bars). However, it is important to note that the experiments in Aksentijevic et al were performed in the mouse heart. The much smaller heart size and smaller n number in our original work may have masked such effects due to worse signal-to-noise and poorer reproducibility. In the ouabain-treated rat heart with blebbistatin (ie no change in contractility), we have reproducibly seen the surprising preservation of PCr when ATP has fallen. This has been repeated by different experimenters in different cohorts. We would therefore suggest that, taking these factors into consideration, the measurements in this current rat study do not contradict the mouse heart measurements in our previously published paper.

While we do not see a discrepancy with our previous study, we do agree that these changes were very unexpected and indeed paradoxical. Like you, we were indeed surprised to see this, and for this reason, we conducted the experiments in this study to ascertain the mechanisms by which Na elevation leads to downstream alterations in mitochondrial metabolism

The authors try to address this issue with performing saturation transfer experiments, but these are, unfortunately, not done correctly, and the conclusions drawn are based on wrong assumptions.

Thank you for raising this issue and we apologise for this error. You are of course entirely correct. Please see specific comments below.

Main issues:

- 1. Saturation of gamma-ATP allows for calculation of the forward rate constant, forward as meaning ATP synthesis, i.e. PCr-to-ATP, not the other way around as*

you suggest; To explain, the drop in PCr signal is caused by the ^{31}P nuclei leaving PCr and being replenished by saturated nuclei (no signal), so it is the rate of ^{31}P nuclei leaving PCr molecule you are measuring; Hence, your results actually suggest stable ATP production via CK, undermining your decreased ATP supply argument.

We thank the reviewer for this important point. We agree that we had presented this in the wrong direction, so we are grateful to the reviewer for this comment. We have corrected this misunderstanding and reanalysed the saturation transfer data in Figure 2. The reaction scheme in Figure 2F has been corrected where selective saturation of the gamma-ATP peak results in a transfer of saturation (equalization of the Boltzmann populations) in the direction of the black arrows in the equilibria from ATP to PCr and Pi. The simultaneous loss of Boltzmann polarization from the unsaturated PCr and Pi pools back to the saturated gamma-ATP pool are shown by coloured arrows in the equilibria, dark and light blue and arrows, respectively. The analysis of the loss of PCr following gamma-ATP saturation does indeed suggest stable ATP generation by CK, Figure 2G (this panel is unchanged). See follow on response to Q2 below. This has been corrected and the figures redrawn in the revised manuscript. The appropriate changes to the analysis have been made in the corresponding paragraph in Results (see lines 136-150) and Discussion (see lines 330-342).

2. I'm surprised you haven't investigated the Pi-to-ATP reaction, you have all data for it and this is potentially much more exciting for your study.

This is an excellent suggestion; we apologise for missing this point in our original analysis. We have now reanalysed the data for the loss of Pi in the same experiment, i.e. during gamma-ATP saturation, new Figure 2H. We are pleased to report that the flux measured in this experiment, i.e. Pi-to-ATP corresponding to net ATP synthesis is indeed slower under conditions of ouabain induced Na elevation compared to baseline and that this recovers on ouabain washout and also looks to be inhibited by CGP, although the latter result did not reach significance, likely due to the experimental variability when measuring the loss of Pi, which is already a very small peak, see ^{31}P spectra in Figure 2E.

The new Figure 2 is shown below. The ^{31}P spectrum in the PCr saturation transfer experiment in our previous Figure 2E has been removed, the schematic in Figure 2F has been corrected, Figure 2G remains unchanged and Figure 2H shows our new analysis of the loss of Pi in the gamma-ATP saturation transfer experiment.

3. Saturation of PCr alone is not sufficient to explore ATP hydrolysis rates since there is always the $ATP \leftrightarrow Pi + ADP$ reaction, and only through simultaneous saturation of both PCr and Pi the ATP hydrolysis can be estimated;

Agreed PCr saturation alone is not sufficient to explore ATP-to-PCr regeneration as there is always the ATP hydrolysis reaction taking place and therefore the measurement of the loss of gamma-ATP on PCr saturation is a composite of both fluxes. We have removed these data from Figure 2 and the associated text. We did not perform the simultaneous saturation experiment.

4. It is unclear what is the "control" saturation experiment supposed to be controlling, I can presume it is supposed to work as a control for the pointless PCr saturation experiment where gamma-ATP is saturated. You might not need a control experiment at all, if your pulses are selective enough, but if you do, you need to place them mirrored around the exchange partner, i.e. in this case PCr;

The "control" saturation experiment is identical in all respects except the saturation pulse is placed in a blank part of the spectrum to ensure the RF saturation is selective enough, as pointed out by the reviewer, and therefore effectively acts as the unsaturated experiment. We cannot place the RF carrier on the other side of the PCr peak as this is where Pi is located. Therefore, we chose the control saturation to be midway between the alpha and gamma-ATP peaks. From Figure 2E top spectrum it can readily be appreciated that the control saturation is indeed selective enough to not hit either of the alpha or gamma ATP resonances. This has now been clarified both

in the Methods section (see lines 495-500) as well as Figure 2 legend (see lines 758-768).

5. *Can you assume stable apparent T1 across the different hearts during Na infusion and wash-out? Since this has a significant effect on the exchange rate/flux, this would need to be justified;*

If T1 changed because of elevated Na or washout then this could indeed influence the exchange rate/flux measurement. We have no reason to believe that the ^{31}P T1 is Na dependent since the sodium cation is diamagnetic. Indeed, to our knowledge there is no evidence for this in the literature either. We note that Na is also elevated in the CGP experiment where the Na dependent changes observed by ^{31}P NMR are inhibited. This would not be the case if it was a T1 effect.

6. *In relation to your static spectra, I'm not sure you can call your data fully relaxed if cardiac PCr in mice at 9.4T is $2.54 \text{ s} \pm 0.41 \text{ s}$ (10.1002/nbm.3371) and Pi is likely longer. This would suggest you can't directly use your signals for the ΔGATP calculation as described in the referenced paper. You would need to apply a saturation correction first. In addition, the assumption of 20 mM free creatine seems too low (doi: 10.1016/0167-4889(92)90057-i) and since you are not actually calculating concentrations of your ^{31}P metabolites, it is difficult to put into perspective with your data.*

We apologize for not describing our acquisition in sufficient detail. For the ^{31}P experiments the excitation pulse is 60° not 90° therefore we only excite 50% of the magnetization in each scan ($\cos(60^\circ) = 0.5$). Therefore, the magnetization need only recover 50% of its total to be fully relaxed. Our total acquisition time = $d1 + \text{AQ} = 3.85\text{s}$. Assuming a $T1 = 2.54\text{s}$ then the z magnetization will have recovered $1 - \exp(-3.85/2.54) = 80\%$ of its starting value. Since we only excite 50% of the magnetization then we calculate that magnetization would be 90% of its full equilibrium value at the end of each scan. This may introduce a small systematic error however we concur that it is *almost* fully relaxed but not strictly fully relaxed. To avoid confusion, we have removed this statement and specified a 60° flip angle (see line 488).

Regarding the assumption of 20 mM free creatine, we apologise for this misunderstanding, it was not clearly described. In fact, we measured creatine concentration independently in every heart from the ^1H NMR spectrum of heart tissue extracted at the end of each experiment. Averaging the measured $[\text{Cr}]$ across all hearts gives 15.7 mM and was not significantly different between groups. Therefore, somewhat lower than we had previously quoted. We have corrected this oversight in the revised manuscript (see lines 494-495). This does not affect the results previously presented.

We note the previously published value measured by HPLC is $40.3 \pm 2.38 \mu\text{mol/g}$ dry wt (DOI: 10.1016/0167-4889(92)90057-i). Converting to a concentration using the widely reported value for intracellular volume as a function of dry weight of 2.4 ml/g dry wt yields an estimated value of $[\text{Cr}]_i = 16.8 \text{ mM}$. We are happy to see that this is almost exactly in agreement with our values measured by ^1H NMR.

Regarding absolute quantification of concentrations of metabolites from the ^{31}P data, we did not include a reference compound in our measurements since ΔG can be calculated from the ratio of Pi/PCr which can be measured directly from the ratio of peak integrals in the spectrum.

7. While beta-ATP is indeed technically the best to use for ATP quantification, but given its resonance frequency (dependent on Mg concentration and pH and requiring large excitation bandwidth) and the fact it is a triplet, it might be less reliable to fit than the other two peaks. I would suggest double checking the unexpected results by analysing gamma ATP and/or an average of all three peaks.

We do not fit the peaks in the spectrum. We simply perform a baseline correction and then perform peak integration and normalise to the first spectrum. We take the point about excitation bandwidth but we note that the beta-ATP peak is about 1500 Hz from our RF carrier frequency while the RF amplitude is about 3 kHz for an 80ms excitation pulse. The beta-ATP peak is well within the bandwidth of the pulse.

The decreased ATP in the absence of altered PCr is observed in all 3 ATP peaks. For illustration we show representative spectra below for a time matched control heart acquired at 40 min (blue spectrum) and for a ouabain treated heart acquired at 40 min when Na is elevated (red spectrum), displayed with the same vertical scaling. All three ATP peaks have decreased with respect to PCr which is unchanged.

Reviewer #4:

The effect of Na on the Ca-mediated regulation of mitochondrial bioenergetics is already known. In this study, however, the authors do provide a broader view of the cell wide consequences related to the effect of high sodium on cardiomyocyte bioenergetics in general. This was achieved by branching out from its effects on mitochondria and demonstrating how the Na-mediated depletion of mitochondrial calcium may cause "Metabolic Gridlock" characterized by decreased OXPHOS and increased ROS production, which leads to the activation of glycolysis, potentially through HIF-1 stabilization.

We agree that, to some extent, the broad effects of Na on Ca-mediated regulation of mitochondrial bioenergetics have been reported before. Indeed, we would hope that we have contributed to this literature and have appropriately reviewed it in our *Introduction* as this forms the foundation on which these studies are built. We suggest that the novelty of our present study is to show for the first time that these effects are acutely reversible (a very important observation if this is to be a therapeutic target in heart failure) and we also provide mechanistic details of how the downstream consequences are mediated through effects on Complexes I, II, III and CoQ recycling. We also show for the first time that Na elevation results a shift in Gibbs free energy for ATP hydrolysis, rate of ATP hydrolysis, as well as the pseudo-hypoxic stabilisation of HIF1 α . However, in such an acute model there is no reason that gene transcription would be altered on the timescale of our experiments. Consequently, we would suggest that alterations in glycolytic flux will not be driven by HIF-1 α -activated transcription on this timescale. If anything, our model would cause a decrease in cytosolic ATP demand by inhibiting NKA demand for ATP with ouabain, under conditions that contractile ATP demand is unaltered. As an aside, we concur that elevated cytosolic Na may lead to elevated Ca which would activate Ca-ATPases which will have the opposing effect and increase ATP demand. In our present study we have, however, excluded this as a mechanistic explanation as Ca elevation alone, does not recapitulate our metabolic reprogramming (see Figure 5).

We therefore propose that the shift in glycolysis that we observe is not transcriptionally driven. The build-up of lactate is a consequence of inhibition of PDH (a key enzyme whose activity is regulated by mitochondrial matrix Ca) which would cause a backing up of glycolysis even under conditions where glycolytic flux was unaltered. "Metabolic Gridlock" is a good description of our data.

However, this broader view is limited given the authors did not fully interrogate the metabolic/cell wide consequences of this pathological mechanisms.

We completely agree. To comprehensively interrogate all of the downstream cellular consequences is, however, a very big ask. We would argue, however, that we already have significantly contributed to defining the broader "metabolic/cell wide consequences". These include –

- a shift in substrate preference towards glucose and away from fatty acid metabolism (see Aksenjijevic et al).
- a reduced OxPhos mediated ATP supply paradoxically reducing ATP in the maintained presence of PCr, and reduced rate of ATP hydrolysis.

- Cytoplasmic succinate accumulation.
- Increased ROS production.
- Demonstrating that this Na-induced ROS production has a real functional consequence in limiting functional recovery on Na washout.
- Stabilisation of HIF1a - which is presumed to have longer-term down-stream transcriptional implications (see comments below in reference to this).

We fully agree that it would be useful to go on to show how these observations impact broader downstream cell function but this is well beyond the aims and scope of this study and would generate an unmanageably large and unfocussed manuscript. We would further stress that our aims in the current study were to examine the acute mechanisms by which Na alters metabolism and thereby better understand the causal factors that could be targeted therapeutically. In this context, developing therapeutics that target the downstream consequences of these acute changes would not necessarily be beneficial. Therefore, interrogation of the metabolic/cell wide consequences of the pathological mechanisms would muddy the waters regarding the acute mechanisms that we describe here and hope to target mechanistically in our future work.

It would be useful to interrogate these pathways further using cell lines treated with the Na/Ca exchanger inhibitor CGP or a CRISPR knockout. This would allow an in-depth assessment of the metabolic changes that occur to glycolysis and in mitochondria.

We completely agree. Indeed, we have included studies in the current manuscript in isolated cells measuring the effects of Na elevation on O₂ consumption (see Figure 4) and ROS generation (see Figure 6) in isolated cardiomyocytes measured by Seahorse XFe24 assays and MitoSOX, respectively, including the effects of the Na/Ca exchange inhibitor CGP. Chronic NCLX knockout is acutely lethal (Luongo: doi: 10.1038/nature22082) and has been shown to promote heart failure; it is unclear what CRISPR knockout in a cell line would add in addition to the use of CGP? However, we agree that more can be done in cell-based models (see specific responses below).

Additionally, such a model would allow for the interrogation of mitochondrial ROS generation and the amount formed during RET (assuming that RET is activated in this model, which is in consideration of the fact that mitochondria contain 12 ROS generators).

Indeed, we have interrogated ROS generation in isolated cells as suggested, see previous comment. However, the referee points out the complexity of answering whether it is due to RET and highlights the 12 ROS generators contained in the mitochondria. These questions, at least in part, are being explored in current and future work but is well beyond the scope of the current manuscript, thus not discussed in the manuscript.

An assessment of HIF-1 activation would also be allowed through transcriptome analyses. Other transcription factors can also be measured like NRF2, HSF-1, and NFκB.

To our knowledge acute Na driven stabilisation of HIF-1 α even in the absence of hypoxia has never been reported in the literature, this is therefore an important finding. However, the chronic downstream consequences of HIF1 α stabilisation have been extensively reported in the hypoxia literature. We agree that this is very important and interesting in the context of heart failure. Our present study, however, is limited to the acute effects of Na elevation (over minutes), its reversibility and acute metabolic reprogramming. It seems unlikely that gene transcription changes would be detectable within this short <1 h timeframe. Future studies in chronic models of Na elevation and heart failure would be required to interrogate gene transcription. We agree that other transcription factors can also be measured, like NRF2, HSF-1, and NF κ B. However, at present we do not have any mechanistic basis to hypothesize that there would be acute Na driven changes in these transcription factors within the very short acute timeframe of these studies.

Thus, I believe this study has a lot of merit and could be suitable for Nature Communications. But it falls short by missing the “big picture” in terms of the metabolic adaptations that are invoked in response to the Na overload and the pathological consequences that follow. Many more experiments would be required.

We are grateful for the reviewer's helpful comments. We agree and indeed many more experiments are planned. However, in the context of the present study, we would like to stress our previous comments that our aim was to interrogate the acute response to Na elevation and not the chronic pathological down-stream consequences of, for example, limiting ATP supply and HIF1 α stabilisation, which are, of course, many and complex. Such wide-spread studies would be muddied by the plethora of competing biochemical mechanisms that may be the outcome of, and not related to, the causal underlying factors. Such studies would not help define the causal mechanisms that may be targeted therapeutically.

REVIEWERS' COMMENTS

Reviewer #2 (Remarks to the Author):

The authors have been responsive and adequately addressed. All the points that I raised in my review and I have no further comments. I think this work makes a significant contribution to this field.

Minor point: The authors added text at the end of the Discussion that included Ref #35 and 36, but those are missing from the reference list (which ends at #34).

Reviewer #3 (Remarks to the Author):

I would like to thank the authors for incorporating the important changes in their saturation transfer experiments, which now nicely pair their lower Pi-to-ATP reaction rate to the decreased oxygen consumption. I have only few minor comments left.

- 1) The decrease in Pi-to-ATP rate is fascinating, but in the light of increased Pi concentration the total flux might actually be preserved. This would be interesting to see, if it is the case.
- 2) The control saturation experiment is performed to make sure your saturation pulse is not directly saturating your exchange partner. Hence you have to mirror it around the PCr/Pi to actually account for the possibility of direct saturation. Still, looking at the spectra in Figure 2 it is likely that the saturation is narrow enough, so please take this more as a recommendation for future.
- 3) With a 60° excitation you recover ~90% of PCr magnetization, however your Pi is likely to be saturated more due to longer T1. This will indeed introduce a small systematic error to your absolute values. The comparison between groups (assuming stable T1) will still be valid, but comparison to literature might prove tricky.
- 4) The new merged version of the manuscript seems to be missing references (only 34 displayed, while count in the manuscript goes up to 41)
- 5) Your figure 2 caption needs updating. Panels G and H are referenced as F and G.

Reviewer #4 (Remarks to the Author):

The author responses to my concerns were adequate. Really nice study.

Responses to Editor and Reviewers' comments.

We thank the editors and the reviewers for their swift and constructive review process and particularly to Reviewer 3 for further insightful comments. We have now revised the manuscript addressing all the issues raised. A detailed itemised response is provided below. We have provided the requested information in the form of the graphs shown below in the Data Supplement. Each original comment is provided in *red italics* followed by our response. Further corrections in the manuscript itself are now highlighted in **green**, in addition to those previously highlighted in **yellow**. We have also edited the Abstract to be under 150 words.

Reviewer #3:

I would like to thank the authors for incorporating the important changes in their saturation transfer experiments, which now nicely pair their lower Pi-to-ATP reaction rate to the decreased oxygen consumption. I have only few minor comments left.

Thank you for your supportive comments.

1) The decrease in Pi-to-ATP rate is fascinating, but in the light of increased Pi concentration the total flux might actually preserved. This would be interesting to see, if it is the case.

Thank you for this interesting suggestion. Indeed, if magnetization transfer is lower, as we have measured, but Pi is higher, as we have also measured, then the overall rate given by the product of the rate constant x concentration (peak area) could indeed be conserved. To answer this question, we have revisited our original data. The below figure recapitulates our Figure 2h of the revised manuscript.

The new analysis is shown below where the total flux has now been calculated by multiplying the magnetization transfer ΔM_z by Pi (expressed relative to baseline). We now included this graph and brief discussion (as above) in the Data Supplement.

We still find a decrease of the total flux under conditions of high Na and interestingly a sustained reduction during washout, thus recapitulating our findings for ATP during high Na and washout in Figure 2a. This is not observed for the time matched control nor for the CGP treated hearts. Thus, everything points towards a reduction in ATP synthesis and this new analysis does not change our original conclusion which we have not changed in the revised manuscript.

We have inserted the statement (lines 134-136):

We also assessed the net flux for ATP synthesis given by $\Delta M_z \times [Pi]$ and found this to be similarly decreased by elevated Na (data not shown). We also included this graph and brief discussion (as above) in the Data Supplement.

2) The control saturation experiment is performed to make sure your saturation pulse is not directly saturating your exchange partner. Hence you have to mirror it around the PCr/Pi to actually account for the possibility of direct saturation. Still, looking at the spectra in Figure 2 it is likely that the saturation is narrow enough, so please take this more as a recommendation for future.

Yes, we were satisfied that the saturation pulse was indeed selective enough to not directly saturate the exchange partner. But we take your recommendation for future experiments.

3) With a 60° excitation you recover ~90% of PCr magnetization, however your Pi is likely to be saturated more due to longer T1. This will indeed introduce a small systematic error to your absolute values. The comparison between groups (assuming stable T1) will still be valid, but comparison to literature might prove tricky.

Yes, we concur that this could introduce a small systematic error in our calculation of ΔG . However, we would point out that ΔG is calculated from the ratio of Pi/PCr. The ratio is less sensitive to such errors if both PCr and Pi are slightly underestimated. Our ΔG values in Figure 2d are in keeping with the literature and therefore comparisons can be made. Regarding the individual measurements of ATP, PCr and Pi in Figure 2a-c these have all been normalized to their respective baselines which will be independent of any small saturation effects due to T1.

4) The new merged version of the manuscript seems to be missing references (only 34 displayed, while count in the manuscript goes up to 41)

We apologize for the missing references and misnumbering some of them. This has been corrected. There are 39 references in total, we have added refs 35-39 which were absent. References 40 and 41 were misnumbered and have now been corrected.

5) Your figure 2 caption needs updating. Panels G and H are referenced as F and G.

Thank you, this has been corrected.